# Is Fairness Only Metric Deep? Evaluating and Addressing Subgroup Gaps in DML

**Natalie Dullerud**[1], **Karsten Roth**[2], **Kimia Hamidieh**[1], **Nicolas Papernot**[1], **Marzyeh Ghassemi**[3]
[1]University of Toronto & Vector Institute, [2]University of Tübingen, [3]MIT

## Abstract

Deep metric learning (DML) enables learning with less supervision through its emphasis on the similarity structure of representations. There has been much work on improving generalization of DML in settings like zero-shot retrieval, but little is known about its implications for fairness. In this paper, we are the first to evaluate state-of-the-art DML methods trained on imbalanced data, and to show the negative impact these representations have on minority subgroup performance when used for downstream tasks. In this work, we first define fairness in DML through an analysis of three properties of the representation space – inter-class alignment, intra-class alignment, and uniformity – and propose *finDML*, the *f*airness *i*n *n*on-balanced *DML* benchmark to characterize representation fairness. Utilizing *finDML*, we find bias in DML representations to propagate to common downstream classification tasks. Surprisingly, this bias is propagated even when training data in the downstream task is re-balanced. To address this problem, we present Partial Attribute De-correlation (*PARADE*) to de-correlate feature representations from sensitive attributes and reduce performance gaps between subgroups in both embedding space and downstream metrics.

## 1 Introduction

Deep metric learning (DML) extends standard metric learning to deep neural networks, where the goal is to learn metric spaces such that embedded data sample distance is connected to actual semantic similarities (Globerson & Roweis, 2006; Weinberger et al., 2006; Hoffer & Ailon, 2018; Wang et al., 2014). The explicit optimization of similarity makes deep metric spaces well suited for usage in unseen classes, such as zero-shot image or video retrieval or facial re-identification (Milbich et al., 2021; Roth et al., 2020c; Musgrave et al., 2020; Hoffer & Ailon, 2018; Wang et al., 2014; Schroff et al., 2015; Wu et al., 2018; Roth et al., 2020c; Brattoli et al., 2020; Hu et al., 2014; Deng et al., 2019; Liu et al., 2017). However, while DML is effective in establishing notions of similarity, work describing potential fairness issues is limited to individual fairness in standard metric learning (Ilvento, 2020), disregarding embedding models.

Indeed, the impacts and metrics of fairness are well studied in machine learning (ML) generally, and representation learning specifically (Dwork et al., 2012; Mehrabi et al., 2019; Locatello et al., 2019b). This is especially true on high-risk tasks such as facial recognition and judicial decision-making (Chouldechova, 2017; Berk, 2017), where there are known risks to minoritized subgroups (Samadi et al., 2018). Yet, relatively little work has been done in the domain of DML (Rosenberg et al., 2021). It is crucial to address this knowledge gap – if DML embeddings are used to create *upstream* embeddings that facilitate *downstream* transfer tasks, biases may propagate unknowingly.

To tackle this issue, this work first proposes a benchmark to characterize *f*airness *i*n *n*on-balanced *DML* - *finDML*. *finDML* introduces three subgroup fairness definitions based on feature space performance metrics – recall@k, alignment and group uniformity. These metrics measure clustering ability and generalization performance via feature space uniformity. Thus, we select the metrics for our definitions to enforce independence between inclusion in a particular cluster or class, and a protected attribute (given the ground-truth label). We leverage existing datasets with fairness limitations (CelebA (Liu et al., 2015) and LFW (Huang et al., 2007)) and induce imbalance in training data of standard DML benchmarks, CARS196 (Krause et al., 2013) and CUB200 (Wah et al., 2011), in order to create an effective benchmark for fairness analysis in DML.

Making use of *finDML*, we then perform an evaluation of 11 state-of-the-art (SOTA) DML methods representing frequently used losses and sampling strategies, including: ranking-based losses (Wang et al., 2014; Hoffer & Ailon, 2018), proxy-based (Kim et al., 2020) losses, semi-hard sampling (Schroff et al., 2015) and distance-weighted sampling (Wu et al., 2018). Our experiments suggest that imbalanced data during upstream embedding impacts the fairness of all benchmarks methods in both upstream embeddings (subgroup gaps up to 21%) as well as downstream classifications (subgroup gaps up to 45.9%). **This imbalance is significant even when downstream classifiers are given access to balanced training data, indicating that data cannot naively be used to de-bias downstream classifiers from imbalanced embeddings.**

Finally, inspired by prior work in DML on multi-feature learning (Milbich et al., 2020), we introduce PARtial Attribute DE-correlation (PARADE). PARADE addresses imbalance by de-correlating two learned embeddings: one learnt to represent similarity in class labels, and one learnt to represent similarity in the values of a sensitive attribute, which is discarded at test-time. This creates a model in which the ultimate target class embeddings have been de-correlated from the sensitive attributes of the input. We note that as opposed to previous work on variational latent spaces, PARADE de-correlates a learned similarity metric. We find that PARADE reduces gaps of SOTA DML methods by up to 2% downstream in *finDML*.

In total, our contributions can be summarized as follows:

1. We define ***finDML***; introducing three definitions of fairness in DML to capture multi-faceted minoritized subgroup performance in upstream embeddings through focus on feature representation characteristics across subgroups, and five datasets for benchmarking.
2. We analyze SOTA DML methods using *finDML*, and find that common DML approaches are significantly impacted by imbalanced data. We show empirically that learned embedding bias cannot be overcome by naive inclusion of balanced data in downstream classifiers.
3. We present ***PARADE***, a novel adaptation of previous zero-shot generalization techniques to enhance fairness guarantees through de-correlation of class discriminative features with sensitive attributes.

## 2 BACKGROUND

**Deep Metric Learning** DML extends standard metric learning by fusing feature extraction and learning a parametrized metric space into one end-to-end learnable setup. In this setting, a large convolutional network $\psi$ provides the mapping to a feature space $\Psi$, while a small network $f$, usually a single linear layer, generates the final mapping to the metric or embedding space $\Phi$. The overall mapping from the image space $X$ is thus given by $\phi = f \circ \psi$. Generally, the embedding space is projected on the unit hypersphere $\mathcal{S}^{D-1}$ through normalization (Weisstein, 2002; Wu et al., 2018; Roth et al., 2020c; Wang & Isola, 2020) to limit the volume of the representation space with increasing embedding dimensionality. The embedding network $\phi$ is then trained to provide a metric space $\Phi$ that operates well under some predefined, usually non-parametric metric such as the Euclidean or cosine distance defined over $\Phi$.

Typical objectives used to learn such metric spaces range from contrastive ranking-based training using tuples of data, such as pairwise (Hadsell et al., 2006), triplet- (Schroff et al., 2015; Wu et al., 2018) or higher-order tuple-based training (Sohn, 2016; Wang et al., 2020a), procedures to bring down the effective complexity of the tuple space (Schroff et al., 2015; Harwood et al., 2017; Wu et al., 2018) or the introduction of learnable tuple constituents (Movshovitz-Attias et al., 2017; Qian et al., 2019; Kim et al., 2020).

More recent work (Milbich et al., 2020; Roth et al., 2020c; Jacob et al., 2019) extends standard DML training through incorporation of objectives going beyond just sole class label discrimination: e.g., through the introduction of artificial samples (Lin et al., 2018; Duan et al., 2018), regularization of higher-order moments (Jacob et al., 2019), curriculum learning (Zheng et al., 2019; Harwood et al., 2017; Roth et al., 2020a), knowledge distillation (Roth et al., 2020b) or the inclusion of additional features (DiVA) to produce diverse and de-correlated representations (Milbich et al., 2020).

**DML Evaluation** Standard performance measures reflect the goal of DML: namely, optimizing an embedding space $\Phi$ for best transfer to new test classes via learning semantic similarities. As

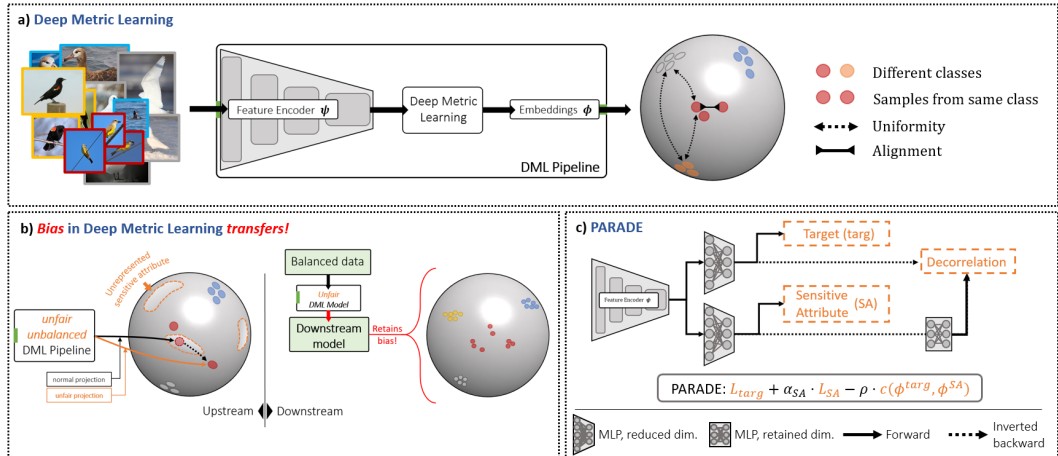

Figure 1: **a)** Visualization of the standard DML pipelines and the aspects of intra-class alignment and uniformity in the embedding space. **b)** Infographic of the fairness issue in DML, where learned representational bias can even transfer to downstream models building on previously learned representations. **c)** Layout of our proposed PARADE approach to better incorporate sensitive attribute context and improve representational fairness.

immediate applications are commonly found in zero-shot clustering or image retrieval, respective retrieval and clustering metrics are predominantly utilized for evaluation. Recall@k (Jegou et al., 2011) or mean average precision measured on recall (Roth et al., 2020c; Musgrave et al., 2020) typically estimate retrieval performance. Normalized mutual information (NMI) on clustered embeddings (Manning et al., 2010) is used as a proxy for clustering quality (see Supplemental for detailed definitions). We leverage these performance metrics to inform *finDML* and our experiments.

**Fairness in Classification**   Formalizing fairness in ML continues to be an open problem (Mehrabi et al., 2019; Chen et al., 2018a; Chouldechova, 2017; Berk, 2017; Locatello et al., 2019b; Chouldechova & Roth, 2018; Dwork et al., 2012; Hardt et al., 2016; Zafar et al., 2017). In classification, definitions for fairness such as demographic parity, equalized odds, and equality of opportunity, rely on model outputs across the random variables of protected attribute and ground-truth label (Dwork et al., 2012; Hardt et al., 2016).

**Fairness in Representations**   A more relevant family of fairness definitions for DML would be those explored in fairness for general representation learning (Edwards & Storkey, 2015; Beutel et al., 2017; Louizos et al., 2015; Madras et al., 2018). Here, the goal is to learn a *fair* mapping from an original domain to a latent domain so that classifiers trained on these representations are more likely to be agnostic to the sensitive attribute in unknown downstream tasks. This assumption distinguishes our setting from previous fairness work in which the downstream tasks are known at train time (Madras et al., 2018; Edwards & Storkey, 2015; Moyer et al., 2018; Song et al., 2019; Jaiswal et al., 2019). DML differs from this form of representation learning as it aims to learn a mapping capturing semantic similarity, as opposed to latent space representation.

Earlier works in fair representation learning intended to obfuscate *any* information about sensitive attributes to approximately satisfy demographic parity (Zemel et al., 2013) while a wealth of more recent works focus on using adversarial methods or feature disentanglement in latent spaces of VAEs (Locatello et al., 2019a; Kingma & Welling, 2013; Gretton et al., 2006; Louizos et al., 2015; Amini et al., 2019; Alemi et al., 2018; Burgess et al., 2018; Chen et al., 2018b; Kim & Mnih, 2018; Esmaeili et al., 2019; Song et al., 2019; Gitiaux & Rangwala, 2021; Rodríguez-Gálvez et al., 2020; Sarhan et al., 2020; Paul & Burlina, 2021; Chakraborty et al., 2020). In this setting, the literature has focused on optimizing on approximations of the mutual information between representations and sensitive attributes: maximum mean discrepancy (Gretton et al., 2006) for deterministic or variational (Li et al., 2014; Louizos et al., 2015) autoencoders (VAEs); cross-entropy of an adversarial network that predicts sensitive attributes from the representations (Edwards & Storkey, 2015; Xie et al., 2017; Beutel et al., 2017; Zhang et al., 2018; Madras et al., 2018; Adel et al., 2019; Zhao &

Gordon, 2019; Xu et al., 2018); balanced error rate on both target loss and adversary loss (Zhao et al., 2019); Weak-Conditional InfoNCE for conditional contrastive learning (Tsai et al., 2021).

PARADE shares aspects of these previous methods in its choice of de-correlation or disentanglement. However, PARADE de-correlates the learned similarity metric as opposed to the latent space. In addition, with *DML-specific* criteria, PARADE learns similarities *over the sensitive attribute* while not directly removing *all* information about the sensitive attribute, as the sensitive attribute and target class embeddings share a base network.

## 3 Extending Fairness to DML - *finDML* Benchmark

To characterize fairness with *finDML*, this section introduces the key constituents – definitions to characterize fairness in embedding spaces and respective benchmark datasets.

### 3.1 Preliminaries

Our embedding space fairness definitions rely on embedding space metrics adapted from (Wang & Isola, 2020) and (Roth et al., 2020c), namely alignment and uniformity. Both metrics we use to characterize embeddings for our definitions in the next section (*intra-* as well as *inter-class alignment* and *uniformity*) have been successfully linked to generalization performance in contrastive self-supervised and metric learning models (Wang & Isola, 2020; Roth et al., 2020c; Sinha et al., 2020). Alignment succinctly captures the similarity structure learned by the representation space with respect to the target labels through measuring distances between pairs of samples. On the other hand, notions of uniformity can differ. Uniformity of the *sample distribution over the hypersphere* has been studied through the radial basis function (RBF) over pairs of samples. Alternatively, uniformity of the *feature space* has been studied through the KL-divergence $\mathcal{D}_{\text{KL}}$ between the discrete uniform distribution $\mathcal{U}_D$ and the sorted singular value distribution $\mathcal{S}_{\phi(X)}$ of the representation space $\phi$ on dataset $X$.

$$U_{\text{KL}}(X) = \mathcal{D}_{\text{KL}}\left(\mathcal{U}_{\text{D}}, \mathcal{S}_{\phi(X)}\right) \tag{1}$$

Here, lower scores indicate more significant directions of variance in learned representations. Both introduced notions of uniformity represent important aspects of the embedding space, but the computational overhead in computing RBF over all pairs of samples in large datasets makes it impractical for our uses and is less interpretable than $U_{\text{KL}}$. Therefore, we leave the uniformity metric utilized in *finDML* general, but utilize $U_{\text{KL}}$ for our experiments.

### 3.2 Defining Fairness

Building on the aforementioned performance metrics, we introduce three definitions for fairness in the embedding spaces of DML models. As the recall@k and alignment metrics inform inclusion in an embedded cluster (or class), we follow fair classification literature in the motivation for our first fairness definition: inclusion in a class should be independent of a protected attribute *given* the ground-truth label. Thus, we examine the probability of encountering a data instance of the same class in a data point's $k$-nearest neighbors to form the first definition. The second definition relies on equal expectation of alignment across sensitive attribute values. Departing from classification literature, our third definition encapsulates fairness in a task-agnostic sense (as DML is often applied in such settings): fairness across the "goodness" of the learned features via a uniformity metric.

Let $X$ denote the input data, and $A$ a protected attribute variable. Denote $X_a$ the partition of $X$ with attribute $a \in A$. To recap common DML terminology, a *positive* pair of samples is defined as $(x_1, x_2) \in X \times X$ s.t. the class label of $x_1$ and $x_2$ are identical. A *negative* pair of samples is defined as $(x_1, x_2) \in X \times X$ such that the class label of $x_1$ and $x_2$ differ. Let $\mathbb{P}_a$ denote the set of all *positive pairs* s.t. at least one of $x_1$ or $x_2$ has attribute $a \in A$, and analogously for $\mathbb{N}_a$ and *negative pairs*.

**Definition 1** (*K*-Close Fairness). *Define $NN_k : \Phi \subset \mathcal{S}^{D-1} \rightarrow \mathcal{P}(X)$ as a function that receives a point $\phi(x) \in \Phi$ and returns a set in the powerset of $X$, $\mathcal{P}(X)$, containing points in $X$ that map to the $k$ nearest neighbors of $\phi(x)$ in $\Phi$. Thus, $\phi$ is $k$-close fair with respect to attribute $A$ if:*

$$\Pr_{x \in X_a}\left(\exists \tilde{x} \in NN_k(\phi(x)) s.t. Y(\tilde{x}) = Y(x)\right) = \Pr_{x \in X_b}\left(\exists \tilde{x} \in NN_k(\phi(x)) s.t. Y(\tilde{x}) = Y(x)\right) \quad \forall a, b \in A \tag{2}$$

*Note: the criteria weakens as $k$ increases, similar to recall@k.*

**Definition 2** (Alignment). *$\phi$ is fair, according to alignment with respect to attribute $A$, if:*

$$\mathbb{E}_{(x_1,x_2)\in\mathbb{P}_a}[||\phi(x_1)-\phi(x_2)||^2] = \mathbb{E}_{(x_1,x_2)\in\mathbb{P}_b}[||\phi(x_1)-\phi(x_2)||^2] \qquad (3)$$

$$\mathbb{E}_{(x_1,x_2)\in\mathbb{N}_a}[||\phi(x_1)-\phi(x_2)||^2] = \mathbb{E}_{(x_1,x_2)\in\mathbb{N}_b}[||\phi(x_1)-\phi(x_2)||^2] \quad \forall a,b\in A \qquad (4)$$

*i.e. the expectation of the alignment is equal across domain of $A$.*

**Definition 3** (Uniformity Across Groups). *$\phi$ is fair, according to uniformity, and with respect to attribute $A$, if the expectation of the uniformity is equal across domain of $A$:*

$$U(\phi(X_a)) = U(\phi(X_b)) \quad \forall a,b\in A \qquad (5)$$

*where $U(\cdot)$ denotes some measure of uniformity over a set $V\in\mathcal{S}^{D-1}$.*

### 3.3 CONSTRUCTED *finDML* BENCHMARK DATASETS

*finDML* encompasses existing DML benchmark datasets, CUB200 and CARS196, and facial recognition datasets, CelebA and LFW (Wah et al., 2011; Krause et al., 2013; Liu et al., 2015; Huang et al., 2007). For fairness analysis, we investigate bird color in CUB200[1], Race in LFW and Skintone in CelebA (Kumar et al., 2009). A detailed description of dataset and attribute labeling is included in the Supplemental. To create additional *fairness* benchmarks, we induce class imbalance in CUB200 and CARS196, as both datasets are naturally balanced w.r.t. class.

**Manually Introduced Class Imbalance** We introduce imbalance by reducing the number of training data samples of 50 randomly selected classes by 90% (Imbalanced). We run an experiment with the original datasets as a balanced control (Balanced) for comparison. In the imbalanced setting, we adjust (increase) the number of training samples of the majoritized groups to match the number of datapoints in the balanced control experiments. We average metrics over 10 sets of 50 randomly selected classes for imbalanced experiments. We use the standard ratio of $50-50$ for train-test split of these datasets, but split over number of data points per class, as opposed to splitting over the classes themselves. The manually imbalanced datasets are used to benchmark standard DML methods, validate our framework, and analyze downstream effects.

Although dataset imbalance does not constitute the sole source of bias in machine learning applications, unfairness as a result of imbalance is the most well-understood in the literature (Chen et al., 2018a). Additionally, we do not assume for our naturally imbalanced datasets, particularly the facial datasets, that attribute imbalance is the *only* source of bias we observe.

## 4 PARTIAL ATTRIBUTE DE-CORRELATION (PARADE)

In this section, we present Partial Attribute De-correlation, or PARADE, in which we incorporate adversarial separation (Milbich et al., 2020) during training to de-correlate separate embeddings. We enumerate several significant changes: 1) only target embedding released at test-time; 2) triplet formation and loss term w.r.t. sensitive attribute; 3) de-correlation with sensitive attribute as opposed to de-correlation to reduce redundancy in concatenated feature space. These two representations branch off from the deep metric embedding model at the last layer. The two representations encode the similarity metrics learned over the sensitive attribute and target class, respectively. The sensitive attribute embedding layer is discarded at test time. The resulting network expresses a similarity metric with respect to the target class, de-correlated from the sensitive attribute (Figure 1). Therefore, PARADE figuratively optimizes the first two fairness definitions proposed in Section 3.2 via an objective that maximizes independence between the sensitive attribute and target class.

**Objective Term Per Embedding** To achieve efficient training and de-correlation of the *target class* and the *sensitive attribute* embedding layers, we simultaneously train both layers that branch from the penultimate layer of the model and de-correlate at each iteration. Because PARADE must learn one embedding w.r.t. target class ($\phi_{targ}$) and one embedding w.r.t. the sensitive attribute ($\phi_{SA}$), we introduce separate objectives for each embedding:

$$\mathcal{L}_{targ} = \frac{1}{N}\sum_{t\sim\mathcal{T}_{targ}}\mathcal{L}(t) \qquad \mathcal{L}_{SA} = \frac{1}{N}\sum_{t\sim\mathcal{T}_{SA}}\mathcal{L}(t)$$

---

[1]While bird color in CUB200 does not represent a real-world fairness setting, CUB200 is widely used as a DML benchmark. Thus, a fairness angle allows fairness analysis of previous methods benchmarked on CUB200.

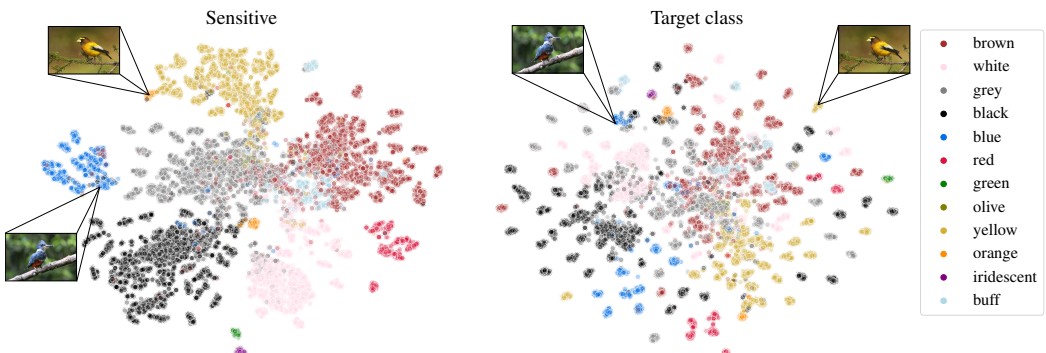

Figure 2: A t-SNE (Maaten & Hinton, 2008) visualization of the two distinct PARADE embeddings for bird color CUB200 experiments: the sensitive attribute embedding (**left**) and the class label embedding (**right**). In the sensitive attribute embedding, both example images are mapped to clusters with birds of the same plumage (yellow and blue, respectively). Due to de-correlation, in the class label embedding, the images are separated from the region of space with other birds of the same plumage, but are still well-clustered, indicating that PARADE can find other attributes to distinguish these species clusters.

where $N$ is the number of training triplet samples, and $\mathcal{L}$ represents a generic loss function, such as triplet loss (Hoffer & Ailon, 2018). We use $t \sim \mathcal{T}_{targ}$ to illustrate sampling over triplets of the form $(x_a, x_p, x_n)$ where $x_a$ and $x_p$ are of the same target class and $x_a$ and $x_n$ are of differing target classes. Similarly, $t \sim \mathcal{T}_{SA}$ indicates sampling over triplets of the form $(x_a, x_p, x_n)$ where $x_a$ and $x_p$ are of the same sensitive attribute subgroup and $x_a$ and $x_n$ are of differing sensitive attribute subgroups. See Figure 2 for a t-SNE visualization of the distinct embeddings of PARADE.

**Partial De-correlation** In order to minimize the correlation between $\phi^{targ}$ and $\phi^{SA}$, we use the adversarial separation (de-correlation) method from (Milbich et al., 2020), which minimizes the mutual information between a pair of embeddings. The task of mutual information minimization is accomplished through learning an MLP to maximize the pair's correlation, $c$, and consequently performing a gradient reversal $R$, which inverts the gradients during backpropagation. The MLP, $\xi$, is trained to maximize $c(\phi_i^{targ}, \phi_i^{SA}) = \|R(\phi_i^{targ}) \odot \xi(R(\phi_i^{SA}))\|_2^2$, s.t. $\odot$ denotes element-wise multiplication. Combining the loss terms results in total loss:

$$\mathcal{L}_{PARADE} = \mathcal{L}_{targ} + \alpha_{SA}\mathcal{L}_{SA} - \rho \cdot c(\phi^{targ}, \phi^{SA})$$

where $\alpha_{SA}$ weights the sensitive attribute loss and $\rho$ weights the degree of de-correlation. $\rho$ modulates the de-correlation term to allow $\psi$ to retain some attribute information (i.e. *partial* de-correlation). Thus, the deployed model $\phi_{targ} = f_{targ} \circ \psi$ can retain information about the sensitive attribute in its feature representations, as $\alpha_{SA}\mathcal{L}_{SA}$ appears in the loss function back-propagated through the full model $\psi$. The extent to which the sensitive attribute affects the output features is controlled by $\alpha_{SA}$; we suggest optimizing $\alpha_{SA} \in (0, 1)$ and $\rho$ through maximization of worst-group performance (Lahoti et al., 2020) (See Supplemental C.5 for further analysis of PARADE hyperparameters).

## 5 EXPERIMENTS

**Baseline DML Methods** For all datasets, we use a ResNet-50 (He et al., 2016) architecture with best performing parameters on a validation set (for further implementation details, see Supplemental). To investigate a sweeping set of frequently used DML methods, we benchmark across a diverse, representative set of 11 techniques, including: three standard ranking-based losses (margin, triplet, n-pair, and contrastive) three batch mining strategies (random, semi-hard and distance-weighted sampling) and three common loss functions (multisimilarity loss, ArcFace loss for handling facial datasets, and proxy-based loss, ProxyNCA) (Hoffer & Ailon, 2018; Hadsell et al., 2006; Wu et al., 2018; Sohn, 2016; Hadsell et al., 2006; Kim et al., 2020; Wang et al., 2020a; Deng et al., 2019; Wu et al., 2018; Schroff et al., 2015). See Supplementary for more details.

**Fairness Evaluation** In the embedding space, we analyze fairness via performance gaps between minoritized groups and majoritized groups, or worst-group performance gaps (Lahoti et al., 2020). For fairness of the feature representations, we compute gaps in three metrics: recall@k and NMI for intra- and inter-class distance (Section 3.2), and the uniformity measure $U_{\mathrm{KL}}$ corresponding to Definition 3 (defined in Section 3.1).

**Training and Evaluation on Downstream Classifiers** To link fairness performance in the embedding space to downstream classification (in which more extensive prior work has been completed), we train downstream classifiers and evaluate classification bias. After training the DML model with the aforementioned criteria, the network is fixed. The output embeddings from the image training datasets, in addition to the class labels, are used to train four downstream classification models: logistic regression (LR), support vector machine (SVM), K-Means (KM), and random forest (RF) (Pedregosa et al., 2011). In the manually imbalanced upstream setting, we train downstream classifiers on the *original balanced image datasets* to ascertain if bias incurred in the embedding can propagate downstream even if the downstream classifier is trained with real balanced data.

We execute class imbalanced experiments for CARS196 and CUB200 and vary the level of imbalance between minoritized and majoritized classes in the upstream training set.

**PARADE Configuration** We test Partial Attribute De-correlation, PARADE, by training models in the listed settings: manually color imbalanced dataset for CUB200, CelebA and LFW. The attribute used to train the sensitive attribute embedding for each dataset, and the attribute used for fairness evaluation. We compare PARADE with margin loss and distance-weighted sampling (Wu et al., 2018) to standard margin loss and distance-weighted sampling.

# 6 RESULTS

## 6.1 SOTA DML METHODS HAVE LARGE FAIRNESS GAPS IN *finDML* BENCHMARK

Our experiments indicate that current DML methods encounter crucial fairness limitations in the presence of imbalanced training data. Table 1 (along with a corresponding table for CARS196 in the Supplemental) demonstrate that gaps in the manually class imbalanced setting are greater than the balanced control setting. In four combinations of loss functions and sampling strategies, we do not observe a scenario in which the class imbalanced setting achieves a smaller gap than the control in the embedding space, *nor* the downstream classification. This is particularly significant due to the nature of sampling strategies studied (Wu et al., 2018; Schroff et al., 2015), which batch samples to force the model to correct "hard" examples. The results validate *finDML* as a benchmark and framework for fairness through the lens of well-studied fairness characterization in classification.

Interestingly, Table 1 displays non-negligible gaps in downstream performance metrics recall and precision even in the balanced control case. This could represent stenography of underlying structures in the data, such as car color or bird size. More likely, however, these gaps are due to use of *macro-*averaging in recall and precision calculations. Nonetheless, the manually class imbalanced settings consistently produce larger gaps.

## 6.2 PROPAGATION OF BIAS TO DOWNSTREAM TASKS

The tabular results emphasize a significant result: naive re-balancing with real data downstream cannot overcome bias incurred in the upstream embedding in any setting studied. Indeed, Table 1 exhibits propagation of bias from upstream embeddings (trained on imbalanced data) to downstream tasks (trained on fixed upstream embeddings with a re-balanced dataset). To provide additional context for the result, we direct to increasing use of DML models as components of larger classification models. This trend is arising in literature such as supervised contrastive learning, and recent developments in pre-training and lifting DML models for classification (Khosla et al., 2020). This necessitates tackling bias in the representation space of DML as opposed to patches downstream, and emphasizes the importance of defining fairness in this setting as done in our work.

**Impact of imbalance degree on lack of fairness** Figure 3 shows that gaps in downstream classification mimic those upstream, even as we vary the level of imbalance introduced when training the upstream embedding. Here, the random forest classifier sees greater gaps in downstream metrics

Table 1: *Gap study on CUB200-2011.* Average gaps in representation space and downstream classification (logistic regressor) over 10 seeds between minoritized and majoritized classes in manually class imbalanced experiments (Imbalanced) and control experiments (Balanced) for CUB200-2011. Results for CARS196 are available in the supplementary with similar conclusions. **Bold** represents larger gap for each method shown (Loss · Batch Mining).

| **Experiments** → | | Balanced | Imbalanced | Balanced | Imbalanced |
|---|---|---|---|---|---|
| **Objective** → | | Margin · Distance | | Margin · Semi-hard | |
| UPSTREAM EMBEDDING | Recall@1 | $0.017 \pm 0.007$ | $\mathbf{0.212 \pm 0.029}$ | $0.02 \pm 0.007$ | $\mathbf{0.187 \pm 0.031}$ |
| | NMI | $-0.001 \pm 0.004$ | $\mathbf{0.112 \pm 0.012}$ | $-0.004 \pm 0.004$ | $\mathbf{0.092 \pm 0.017}$ |
| | $U_{KL}$ | $-0.042 \pm 0.003$ | $\mathbf{0.0 \pm 0.002}$ | $-0.048 \pm 0.004$ | $\mathbf{0.002 \pm 0.004}$ |
| DOWNSTREAM CLASSIFICATION | Precision | $0.339 \pm 0.007$ | $\mathbf{0.39 \pm 0.014}$ | $0.33 \pm 0.004$ | $\mathbf{0.393 \pm 0.015}$ |
| | Recall | $0.36 \pm 0.007$ | $\mathbf{0.424 \pm 0.018}$ | $0.351 \pm 0.005$ | $\mathbf{0.43 \pm 0.016}$ |
| | Accuracy | $0.014 \pm 0.002$ | $\mathbf{0.131 \pm 0.027}$ | $0.016 \pm 0.005$ | $\mathbf{0.131 \pm 0.031}$ |
| **Objective** → | | Triplet · Distance | | Triplet · Semi-hard | |
| UPSTREAM EMBEDDING | Recall@1 | $0.019 \pm 0.006$ | $\mathbf{0.159 \pm 0.031}$ | $0.019 \pm 0.006$ | $\mathbf{0.168 \pm 0.036}$ |
| | NMI | $-0.001 \pm 0.004$ | $\mathbf{0.103 \pm 0.016}$ | $-0.004 \pm 0.006$ | $\mathbf{0.082 \pm 0.016}$ |
| | $U_{KL}$ | $-0.054 \pm 0.006$ | $\mathbf{-0.004 \pm 0.009}$ | $-0.051 \pm 0.006$ | $\mathbf{0.014 \pm 0.011}$ |
| DOWNSTREAM CLASSIFICATION | Precision | $0.336 \pm 0.005$ | $\mathbf{0.41 \pm 0.014}$ | $0.338 \pm 0.007$ | $\mathbf{0.384 \pm 0.014}$ |
| | Recall | $0.357 \pm 0.004$ | $\mathbf{0.459 \pm 0.016}$ | $0.359 \pm 0.007$ | $\mathbf{0.426 \pm 0.016}$ |
| | Accuracy | $0.016 \pm 0.003$ | $\mathbf{0.179 \pm 0.031}$ | $0.02 \pm 0.005$ | $\mathbf{0.134 \pm 0.031}$ |

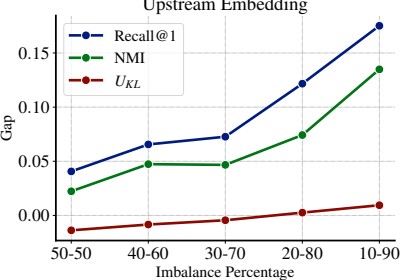
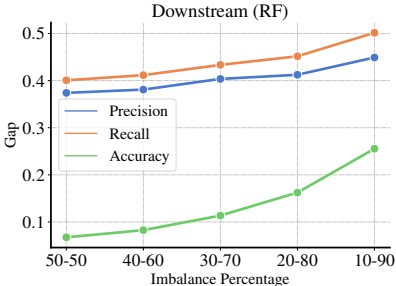

Figure 3: Impact of varying imbalance between the *minoritized* and *majoritized* classes on upstream embedding and downstream classifier (RF) in the manually class imbalanced CARS196 experiments. (Note: the imbalance percentage $50 - 50$ is equivalent to the balanced setting). Gaps increase both upstream and downstream with more imbalance introduced to the upstream training data.

than the control, even when manual imbalance is set at $40 - 60$ upstream, *and* the downstream training dataset is balanced. For results with additional downstream classifiers, see Supplemental. This experiment demonstrates that the propagation of bias to downstream will occur even with lower levels of imbalance, and does not appear to depend on the downstream classifier chosen.

### 6.3 REDUCED SUBGROUP GAPS THROUGH PARTIAL DE-CORRELATION WITH SENSITIVE ATTRIBUTE

Table 2a shows results for performance gaps between relevant subgroups in both facial recognition datasets. PARADE shows strong results for CUB200 bird color dataset, primarily reducing gaps downstream and accordingly to recall@1 (Definition 1). PARADE can reliably reduce gaps for both the representation space and downstream classifiers on LFW. Interestingly, we observe that the majoritized subgroup ("White") had worst performance of all "Race" subgroups (see Supplemental), contrary to previous results (Samadi et al., 2018).[3] As such, we measure gaps between the worst-performing subgroup and others.

---

[2]Due to the great number of singleton classes in LFW, recall@1 is discarded as a metric.

[3]Note: minoritized subgroups can still encounter notable bias across other axes more difficult to measure (Radford & Espenshade, 2014).

Table 2: *Comparison between PARADE and standard losses with distance-weighted sampling* of average gaps in representation space and downstream classification (logistic regressor) over 3 seeds between minoritized and majoritized groups in **(a)** facial dataset studies, namely on CelebA (w.r.t. "Fitzpatrick Skintone") and between worst-performing subgroup and other subgroups in LFW[2](w.r.t. "Race") with *Margin* loss and **(b)** bird color experiments for CUB200 image dataset (w.r.t. color) with *Margin* and *Triplet* loss. **Bold** represents smaller gap (better fairness performance).

(a)

| Facial Datasets | | CelebA (*skintone*) | | LFW (*race*) | |
|---|---|---|---|---|---|
| | | PARADE | Margin · Distance | PARADE | Margin · Distance |
| UPSTREAM EMBEDDING | Recall@1 | **0.085 ± 0.009** | 0.122 ± 0.005 | 0.075 ± 0.014 | **0.068 ± 0.013** |
| | NMI | **−0.012 ± 0.003** | −0.002 ± 0.003 | **0.041 ± 0.003** | 0.048 ± 0.003 |
| | $U_{KL}$ | **−0.04 ± 0.011** | −0.03 ± 0.007 | 0.163 ± 0.003 | 0.165 ± 0.005 |
| DOWNSTREAM CLASSIFICATION | Precision | 0.146 ± 0.006 | **0.1 ± 0.007** | **0.004 ± 0.002** | 0.005 ± 0.005 |
| | Recall | 0.141 ± 0.007 | **0.098 ± 0.007** | **0.003 ± 0.001** | 0.007 ± 0.006 |
| | Accuracy | 0.131 ± 0.006 | **0.082 ± 0.005** | **0.009 ± 0.003** | 0.012 ± 0.009 |

(b)

| CUB200-2011 *color* | | PARADE (M · D) | Margin · Distance | PARADE (T · D) | Triplet · Distance |
|---|---|---|---|---|---|
| UPSTREAM EMBEDDING | Recall@1 | **0.172 ± 0.021** | 0.176 ± 0.041 | **0.172 ± 0.027** | 0.195 ± 0.051 |
| | NMI | 0.349 ± 0.031 | **0.326 ± 0.184** | 0.372 ± 0.291 | **0.359 ± 0.024** |
| | $U_{KL}$ | 0.167 ± 0.013 | **0.153 ± 0.013** | 0.174 ± 0.035 | **0.159 ± 0.018** |
| DOWNSTREAM CLASSIFICATION | Precision | **0.317 ± 0.046** | 0.333 ± 0.049 | **0.248 ± 0.038** | 0.308 ± 0.119 |
| | Recall | **0.352 ± 0.039** | 0.363 ± 0.046 | **0.276 ± 0.042** | 0.337 ± 0.123 |
| | Accuracy | 0.163 ± 0.018 | **0.153 ± 0.028** | **0.148 ± 0.049** | 0.154 ± 0.029 |

For CelebA, we find for standard methods the minoritized subgroups to generally perform worst. PARADE excels at gap reduction upstream but encounters larger subgroup gaps downstream compared to standard methods. While PARADE does reduce downstream gaps between light skintones (I, II, and III), and the two lighter dark skintones (IV, V), gaps increase between lighter skintones and the darkest skintone (VI) (see Supplemental). Because skintone VI constitutes < 1% of the CelebA dataset, PARADE is likely not able to learn similarity between faces over attributes besides skintone. And PARADE is prevented from learning similarities based on skintone due to de-correlation. In such settings, PARADE could be combined with oversampling minoritized subgroups to ensure better performance.

In general, the results show promising benefits of PARADE to adequately address and improve on the challenge of subgroup gaps for DML models used in facial recognition; and in the standard DML dataset CUB200, for recall@1 upstream (Definition 1) and across metrics downstream (Table 2b).

## 7 DISCUSSION

In this work, we introduce the *finDML* benchmark, a framework for fairness in deep metric learning (§3.2). We demonstrate the fairness limitations of established DML techniques, and the surprising propagation of embedding space bias to downstream classifiers. Importantly, we find that this bias cannot be addressed at the level of downstream classifiers but instead needs to be addressed at the DML stage. We investigate the limit of this propagation in manually introduced imbalance, and finally show that PARADE can reduce subgroup gaps in several settings.

**Limitations** PARADE suffers from pitfalls similar to other "fairness with awareness" methods: PARADE uses information only on pre-defined sensitive attributes and therefore can be unfair w.r.t. other sensitive attributes. PARADE does have an advantage in addressing the combinatorial number of attributes considered in multi-attribute fairness through DML, which will scale sub-combinatorially in time/space complexity. We also note that subgroup gaps are not sufficient to capturing societal understandings of fairness, and there is no consensus as to how to remedy such gaps (Chouldechova & Roth, 2018; Dwork et al., 2012; Hardt et al., 2016; Zemel et al., 2013; Zafar et al., 2017). Additionally, while PARADE intentionally optimizes Definitions 1 and 2, we provide no explicit guarantee and optimization of uniformity, Definition 3, remains an open problem. Finally, PARADE does incur slight decrease in overall performance, similar to other methods (Wick et al., 2019) (see Supplemental for per-subgroup performance and additional fairness-utility trade-off analysis for PARADE).

**Code of Ethics Statement** The work presented here deals with fairness in deep metric learning. A portion of our studies in the paper focus on CARS196 and CUB200-2011 datasets, which have consistently been used in benchmarking novel DML frameworks (Krause et al., 2013; Wah et al., 2011). The fairness analysis considered for CUB200-2011 deals with bird color, which does not, to our knowledge, correspond with any societal problems relating to fairness. Nonetheless, as CUB200-2011 is used in a litany of papers for SOTA performance comparison, *finDML* includes CUB200 so that DML methods can be analyzed w.r.t. fairness on a dataset used in their original paper.

We do include facial recognition datasets and tasks and analyze fairness with respect to facial attributes. Facial recognition does raise ethical concerns in practice. We note that our paper attempts to address primary social concerns in facial identity recognition. We do not encourage the task of facial *attribute* recognition, and solely use labeled attributes that correspond to known axes of bias for fairness analysis (e.g. Race and Skintone). As PARADE has solely been tested in two widely used public facial recognition datasets, we cannot guarantee fairness nor privacy in practical settings with private facial datasets.

**Reproducibility Statement** Additional experimental results discussed in the main paper and others are contained in Supplemental C. Implementation details including attribute information, generation of attributes, training parameters, metric calculation and gap computation are listed in Supplemental D. Code available here: `https://github.com/ndullerud/dml-fairness`.

## ACKNOWLEDGEMENTS

We would like to acknowledge and thank our sponsors, who support our research with financial and in-kind contributions: CIFAR through the Canada CIFAR AI Chair, Intel, NFRF through an Exploration grant, NSERC through the COHESA Strategic Alliance, International Max Planck Research School for Intelligent Systems (IMPRS-IS), European Laboratory for Learning and Intelligent Systems (ELLIS) PhD program, Mitacs Accelerate through internship program, and Microsoft Research. Resources used in preparing this research were provided, in part, by the Province of Ontario, the Government of Canada through CIFAR, and companies sponsoring the Vector Institute. We would like to thank members of the CleverHans Lab and HealthyML Lab for their invaluable feedback.

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

# SUPPLEMENTAL MATERIAL

## A  ADDITIONAL BACKGROUND

### A.1  DEEP METRIC LEARNING DEFINITIONS

Here, we iterate through some common DML criteria and batch mining strategies more formally than in the main paper. Throughout this section, let $X$ denote the input data, $\phi(X)$ the embedded data, $Y$ the class label, and let $Y(x)$ denote the value of the ground truth class label for data instance $x$. Denote the set of all positive pairs with respect to class label $Y$ as $\mathbb{P} = \{(x_1, x_2) \in X \times X : Y(x_1) = Y(x_2), x_1 \neq x_2\}$. Denote the set of all negative pairs with respect to class label $Y$ as $\mathbb{N} = \{(x_1, x_2) \in X \times X : Y(x_1) \neq Y(x_2)\}$. We use the notation $(x_a, x_p, x_n) \in X \times X \times X$ to denote a triplet with an anchor sample $x_a$, positive sample $x_p$ where $Y(x_a) = Y(x_p)$, and negative sample $x_n$ where $Y(x_a) \neq Y(x_n)$.

**Batch Sampling and Mining**   The batch sampling procedure in deep metric learning methods differ from that of generic deep classifiers in that canonical loss functions require tuples or pairs of samples in order to utilise ranking objectives as training surrogates to learn an appropriate similarity metric. To ensure that tuples with positive and negative examples can be extracted from the batch, the Samples-Per-Class-$n$ (SPC-$n$) heuristic (see e.g. Roth et al. (2020c)) is generally used, where commonly $n = 2, 4, 8$. Given a batch size $b$, the SPC-$n$ technique randomly selects $b/n$ classes from which $n$ training samples are then drawn randomly to be included in each batch $\mathcal{B}$.

After feeding the batch through the network, tuples are *mined* from the batch to use in the loss function. We refer to mining in this paper either as *batch mining* or overload the both as *batch sampling* terminology. The naive solution to tuple mining is random mining, in which all possible tuples of the form $(x_a, x_p, x_n)$ are considered and $b$ are randomly chosen from the batch. However, this method lacks the capacity to utilize valuable information about the current embedding space, and is prone to significant redundancy in the training signal Schroff et al. (2015); Wu et al. (2018).

**Definition 4** (Random Mining). *Hu et al. (2014) For each $x_a \in \mathcal{B}$, we randomly draw a positive example from $\{x_p \in \mathcal{B} : Y(x_p) = Y(x_a), x_p \neq x_a\}$ and a negative example from $\{x_n \in \mathcal{B} : Y(x_n) \neq Y(x_a)\}$ to form the triplet $(x_a, x_p, x_n)$.*

Intuitively, this could be mitigated by *hard* mining heuristics searching for negative samples that are closer to the anchor sample in the embedding space than positive samples, thereby always ensuring a significant training signal. Unfortunately, such approaches are prone to heavy overfitting, training instability and large gradient variance, thereby commonly resulting in less-than-optimal solutions (see e.g. Schroff et al. (2015); Harwood et al. (2017); Wu et al. (2018)). Recent approaches thus establish more lenient heuristics, such as through the introduction of slack parameters to the hard mining objective (e.g. semi-hard mining Schroff et al. (2015) or softhard mining Roth & Brattoli (2019)).

**Definition 5** (Semi-hard Mining). *For each $x_a \in \mathcal{B}$, we randomly draw a positive example from $\{x_p \in \mathcal{B} : Y(x_p) = Y(x_a), x_p \neq x_a\}$, and a negative example from the set*

$$\{x_n \in \mathcal{B} : Y(x_n) \neq Y(x_a), \|\phi(x_a) - \phi(x_n)\|_2^2 \|\phi(x_a) - \phi(x_p) + \gamma\|_2^2\}$$

*where $\gamma \in \mathbb{R}$ is a slack parameter, to form the triplet $(x_a, x_p, x_n)$.*

While other adaptive means (e.g. Harwood et al. (2017); Roth et al. (2020a)) have shown strong performance improvements, modern predefined heuristics such as distance-weighted tuple mining Wu et al. (2018) offer a better cost-to-performance tradeoff Roth et al. (2020a). Here, the heuristic leverages the fact that embeddings are commonly normalized to have unit $L_2$ norm for regularization purposes Wu et al. (2018). This ensures a distribution over a unit hypersphere, in which explicit pairwise distributions can be established Weisstein (2002); Wu et al. (2018). By inverting this distribution, distance-weighted mining can thus encourage a much more diverse coverage of tuple difficulties, improving generalization performance and reducing gradient variance Wu et al. (2018).

**Definition 6** (Distance-weighted). *For embedding spaces normalized to the $(D-1)$-dimensional hypersphere $\mathcal{S}^{D-1}$, we have Weisstein (2002); Wu et al. (2018) the following pairwise sampling distribution $q(\bullet, \bullet)$:*

$$q\left(d\left(\phi(x_i), \phi(x_j)\right)\right) \propto d\left(\phi(x_i), \phi(x_j)\right)^{D-2}\left[1 - \frac{1}{4}d\left(\phi(x_i), \phi(x_j)\right)\right]^{\frac{D-3}{2}}$$

*for embedding pairs $(\phi(x_i), \phi(x_j)) \in \mathcal{S}^{D-1}$ and Euclidean distance $d(\bullet, \bullet)$. For each $x_a \in \mathcal{B}$, we randomly draw a positive example from $\{x_p \in \mathcal{B} : Y(x_p) = Y(x_a), x_p \neq x_a\}$, and sample a negative example based on an inverse distance distribution w.r.t. $q$:*

$$P(x_n | x_a) \propto min(\lambda, q^{-1}(d(\phi(x_i), \phi(x_j))))$$

*where $\lambda \in \mathbb{R}$ defines a clipping parameter to avoid potentially erroneous training samples.*

**Examined Objectives**    The primary goal of DML loss functions is to provide a training surrogate that implicitly optimizes for desired metric space quantities by narrowing down the expected distance between positive pairs of samples and expanding on the expected distance between negative pairs of samples in the embedding space. Most commonly employed pair Hadsell et al. (2006) and tripled-based Schroff et al. (2015); Hoffer & Ailon (2018) ranking losses penalize close negative pairs and disparate positive pairs up to a predefined margin to avoid overclustering. Using $\mathbb{P}(x)$ to denote all positive pairs containing $x$

$$\mathbb{P}(x) = \{(x_1, x_2) \in \mathbb{P} : x_1 = x\}$$

and $\mathbb{N}(x)$ to denote all negative pairs containing $x$

$$\mathbb{N}(x) = \{(x_1, x_2) \in \mathbb{N} : x_1 = x\}$$

we define

**Definition 7** (Contrastive). *Hadsell et al. (2006) Given a batch $\mathcal{B}$, and pairs of samples $\mathbb{S}$ over $\mathcal{B} \times \mathcal{B}$, the contrastive objective is defined as:*

$$\mathcal{L}_{contr} = \frac{1}{b} \sum_{(x_i, x_j) \in \mathbb{S}} \mathbb{I}_{Y(x_i) = Y(x_j)} d\left(\phi(x_i), \phi(x_j)\right) + \mathbb{I}_{Y(x_i) \neq Y(x_j)} \left[\gamma - d\left(\phi(x_i), \phi(x_j)\right)\right]_+$$

*with margin $\gamma$.*

**Definition 8** (Triplet). *Hoffer & Ailon (2018) The triplet loss extends the contrastive objective with sample triplets and can be defined as:*

$$\mathcal{L}_{tripl} = \frac{1}{b} \sum_{\substack{(x_a, x_p, x_n) \in \mathcal{T} \\ Y(x_a) = Y(x_p) \neq Y(x_n)}} \left[d\left(\phi(x_a), \phi(x_p)\right) - d\left(\phi(x_a), \phi(x_n)\right) + \gamma\right]_+$$

*with margin $\gamma$.*

Margin loss extends the triplet objective through the inclusion of a learnable boundary $\beta$ between positive and negative pairs Wu et al. (2018). In our experiments, we utilise $\beta = 1.2$. These criteria are widely used (see e.g. Roth et al. (2020c); Musgrave et al. (2020)) and require mining to make use of the batch information.

**Definition 9** (Margin). *Wu et al. (2018) The margin objective integrates the learnable distance boundary $\beta$ between positive and negative pairs of samples for a relative ordering of pairs with respect to $\beta$ as*

$$\mathcal{L}_{margin} = \sum_{(x_i, x_j) \in \mathbb{S}} \gamma + \mathbb{I}_{Y(x_i) = Y(x_j)} \left(d\left(\phi(x_i), \phi(x_j)\right) - \beta\right) - \mathbb{I}_{Y(x_i) \neq Y(x_j)} \left(d\left(\phi(x_i), \phi(x_j)\right) - \beta\right)$$

Going beyond pairs and triplets, one can also consider the case of more general n-tuples, which was investigated e.g. in the N-Pair objective Sohn (2016) and the Multisimilarity loss Wang et al. (2020a).

**Definition 10** (N-Pair). *Sohn (2016) N-Pair loss is a simple augmentation of the triplet framework in which all negatives in the batch $\mathcal{B}$ are incorporated in the objective function as:*

$$\mathcal{L}_{npair} = \frac{1}{b} \sum_{\substack{(x_a, x_p) \in \mathcal{B} \\ Y(x_a) = Y(x_p), a \neq p}} \log \left( 1 + \sum_{\substack{x_n \in \mathcal{B} \\ Y(x_a) \neq Y(x_n)}} \exp \left( \phi(x_a)^{*,T} \phi(x_n) - \phi(x_a)^{*,T} \phi(x_p)^* \right) \right) +$$

$$\frac{\nu}{b} \cdot \sum_{i \in \mathcal{B}} \|\phi(x_i)^*\|_2^2 \quad (6)$$

*where $\nu$ denotes an embedding regularization parameter due to slow convergence for normalized embeddings stated in Sohn (2016)*

**Definition 11** (Multisimilarity). *Wang et al. (2020a) Multisimilarity loss fits into the ranking loss category, but in addition to evaluation of cosine similarity between positive-anchor pairs and negative-anchor pairs, the objective evaluates positive-positive and negative-negative pairs with respect to the anchor:*

$$s_c^*(x_i, x_j) = \begin{cases} s_c\left(\phi(x_i), \phi(x_j)\right) & s_c\left(\phi(x_i), \phi(x_j)\right) > \min_{x_j \in \mathbb{P}(x_i)} s_c\left(\phi(x_i), \phi(x_j)\right) - \epsilon \\ s_c\left(\phi(x_i), \phi(x_j)\right) & s_c\left(\phi(x_i), \phi(x_j)\right) < \max_{x_k \in \mathbb{N}(x_i)} s_c\left(\phi(x_i), \phi(x_k)\right) + \epsilon \\ 0 & otherwise \end{cases}$$

$$\mathcal{L}_{multisim} = \frac{1}{b} \sum_{x_i \in \mathcal{B}} \frac{1}{\alpha} \log \left[ 1 + \sum_{x_j \in \mathbb{P}(x_i)} \exp\left(-\alpha \left(s_c^*\left(\phi(x_i), \phi(x_j)\right) - \lambda\right)\right) \right] +$$

$$\frac{1}{\beta} \log \left[ 1 + \sum_{k \in \mathbb{N}(x_i)} \exp\left(\beta \left(s_c^*\left(\phi(x_i), \phi(x_k)\right) - \lambda\right)\right) \right] \quad (7)$$

*where cosine similarity $s_c(x, y) = x^T y$ for two normalized vectors $x, y \in X$.*

Notably, the Multisimilarity loss employs a masking process as a stand-in for the lack of batch-mining heuristic. While this proves to be similarly successfull in addressing the tuple sampling complexity issue, this can also be addressed through the usage of proxy-samples. These are dummy variables that represent various contextual properties (such as mean class representations) to serve as standing for actual samples, which is found e.g. in the ArcFace Deng et al. (2019) or ProxyNCA loss Movshovitz-Attias et al. (2017).

**Definition 12** (Proxy-NCA). *Kim et al. (2020) ProxyNCA learns class proxies, or class centers, which each represent a class in the set of unique classes $\mathcal{Y}$. Then, each anchor from the batch is sampled and a positive or negative proxy $\psi^c \in \mathbb{R}^d$ per class $c \in \mathcal{Y}$ is introduced in lieu of a positive or negative sample, respectively, giving:*

$$\mathcal{L}_{proxy} = -\frac{1}{b} \sum_{x_i \in \mathcal{B}} \log \left( \frac{\exp\left(-d\left(\phi(x_i), \psi_{Y(x_i)}\right)\right)}{\sum_{c \in \mathcal{Y} \setminus \{Y(x_i)\}} \exp\left(-d\left(\phi(x_i), \psi_c\right)\right)} \right)$$

**Definition 13** (Arcface). *Deng et al. (2019) Arcface combines proxy and angular loss methods (e.g. in Wang et al. (2017)) to enforce an angular margin between the embeddings $\phi$ and a proxy (or approximate center) $W \in \mathbb{R}^{c \times d}$ for each class, giving the following:*

$$\mathcal{L}_{arc} = -\frac{1}{b} \sum_{x_i \in \mathcal{B}} \log \frac{\exp\left(s \cdot \cos\left(W_{Y(x_i)}^T \phi(x_i) + \gamma = 0.5\right)\right)}{\exp\left(s \cdot \cos\left(W_{Y(x_i)}^T \phi(x_i) + \gamma = 0.5\right)\right) + \sum_{\substack{x_j \in \mathcal{B} \\ Y(x_i) \neq Y(x_j)}} \exp\left(s \cdot \cos\left(W_{Y(x_j)}^T \phi(x_i)\right)\right)}$$

*where the angular component is encoded in additive angular margin penalty $\gamma$, and $s$ is a scaling parameter, which denotes the radius of the effective utilized hypersphere $\mathcal{S}$.*

**Standard Performance Metrics** Performance metrics in deep metric learning aim to capture the quality of the similarity metric learned by the deep embedding model. Therefore, standard performance metrics in DML reflect the closeness between samples of the same class, the separability

of samples of different classes, the clustering quality of embedding, and the uniformity over the hypersphere embedding space, which has been linked to zero-shot generalization capability Wang & Isola (2020), as discussed in Section 2. In our experiments, we utilize recall@1 Jegou et al. (2011), normalized mutual information Manning et al. (2010) between cluster labels assigned by the well-known K-Means Lloyd (1982) algorithm and ground-truth class labels, and $U_{\mathrm{KL}}$ to measure the closeness between samples of same class, cluster quality of the embedding (and hence, the separability of distinct classes) and uniformity, respectively. Here, we define these metrics formally, but we note that there exist multitudinous performance metrics for DML that we do not define here or use explicitly for our results, including f1 score, mean average precision (mAP), and recall@k for $k > 1$ Jegou et al. (2011).

**Definition 14** (Recall@k). *Jegou et al. (2011) Given $k \in \{1, \dots, |X|\}$, denote $NN_k$ as defined in Definition 1. Then, Recall@k is measured as:*

$$Recall@k = \frac{1}{|X|} \sum_{x \in X} \begin{cases} 1 & \exists \tilde{x} \in NN_k(x) : Y(\tilde{x}) = Y(x) \\ 0 & else \end{cases}$$

**Definition 15** (Normalized Mutual Information Score on Clusters). *Manning et al. (2010) Let $C$ a clustering algorithm, such as K-Means Lloyd (1982) with the number of clusters set to $|Y|$, such that $C(x)$ indicates the cluster label for data point $x \in X$. The normalized mutual information score between the target labels $Y$ and the cluster labels $C$ is measured as:*

$$NMI = \frac{2 \cdot I(Y(X); C(X))}{H(Y(X)) + H(C(X))}$$

*where for random variables $X, Y$, $I(\cdot, \cdot)$ denotes the mutual information function:*

$$I(X; Y) = H(Y) - H(Y|X)$$

*and $H(\cdot)$ denotes the entropy function:*

$$H(X) = - \sum_{x \in X} \Pr(x) \log(\Pr(x))$$

The performance metric $U_{\mathrm{KL}}$, used to measure feature uniformity for our empirical evaluations, is defined in Section 3.1.

## A.2 CLASSIFICATION FAIRNESS DEFINITIONS

Fairness definitions and criteria in classification are briefly mentioned in Section 2 of the main paper. Here, we provide explicit formulas for the most common fairness definitions, including demographic parity, equalized odds, and equality of opportunity Hardt et al. (2016), and provide some additional context on fairness definition evolution.

**Definition 16** (Demographic Parity). *The predictor $\hat{Y}$ satisfies demographic parity with respect to attribute $A$ and class $Y$ if the predictor is independent of $A$:*

$$\Pr[\hat{Y} = 1|A = a] = \Pr[\hat{Y} = 1|A = b] \qquad \forall a, b \in A$$

Specifically, demographic parity has largely been used over the years as a simple and intuitive definition of fairness, in which a classifier is said to satisfy demographic parity if the sensitive attribute is independent of the output of the classifier. While demographic parity provides a simple fairness definition, the measure cannot capture fairness in classification tasks where the ground-truth label is inherently related to a certain attribute value Li et al. (2017).

**Definition 17** (Equalized Odds). *The predictor $\hat{Y}$ satisfies demographic parity with respect to attribute $A$ and class $Y$ if the predictor is independent of $A$ conditional on $Y$:*

$$\Pr[\hat{Y} = 1|A = a, Y = y] = \Pr[\hat{Y} = 1|A = b, Y = y] \qquad \forall a, b \in A, \forall y \in \{0, 1\}$$

*from Hardt et al. (2016).*

**Definition 18** (Equality of Opportunity). *The predictor $\hat{Y}$ satisfies demographic parity with respect to attribute $A$ and class $Y$ if the predictor is independent of $A$ conditional on positively labelled $Y$:*

$$\Pr[\hat{Y} = 1|A = a, Y = 1] = \Pr[\hat{Y} = 1|A = b, Y = 1] \qquad \forall a, b \in A$$

*from Hardt et al. (2016).*

This lead to the introduction of other fairness definitions that capture such nuances, the most well-known of which are probably equalized odds and equality of opportunity Hardt et al. (2016). However, fairness metrics overall have been criticized due to the choice of protected attribute over which to measure, and the inability of these metrics to capture bias with respect to certain attributes which are not known at test-time. We discuss this to a limited extent in Section 7.

# B    DATASET SUMMARY STATISTICS

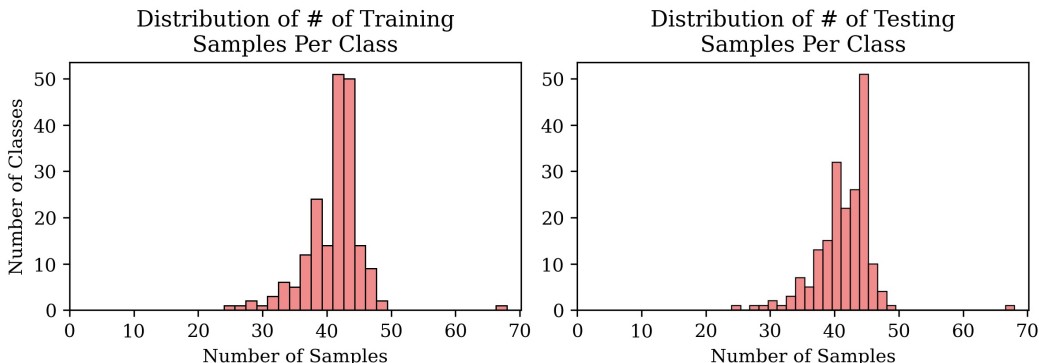

Figure 4: *Class distribution in CARS196.* Histograms visualizing the distribution over number of samples per class in the train (left) and test (right) datasets in CARS196.

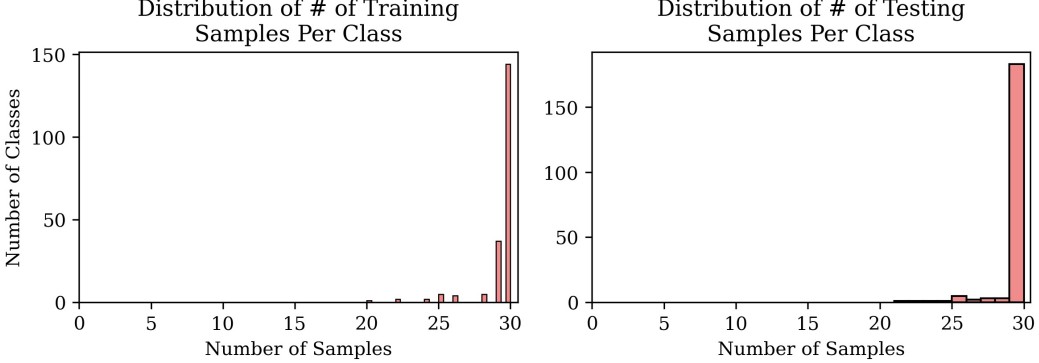

Figure 5: *Class distribution in CUB200.* Histograms visualizing the distribution over number of samples per class in the train (left) and test (right) datasets in CUB200.

|  | Black | Blue | Brown | Buff | Green | Grey | Iridescent | Olive | Orange | Red | White | Yellow |
|---|---|---|---|---|---|---|---|---|---|---|---|---|
| Train | 21.20 | 5.58 | 18.08 | 3.01 | 0.37 | 19.20 | 0.51 | 0.49 | 1.02 | 3.52 | 13.35 | 13.65 |
| Test | 21.17 | 5.56 | 18.11 | 3.04 | 0.39 | 19.21 | 0.51 | 0.51 | 1.01 | 3.52 | 13.29 | 13.68 |

Table 3: *Summary statistics for CUB200 bird color* The percentage of the dataset constituted by each bird color in CUB200, in the train dataset and test dataset, respectively.

# C    ADDITIONAL RESULTS

## C.1    CARS196

Additional results for all loss and batch mining strategies for the manually class imbalanced experiments and balanced controls for CARS196 are located in Tables 6 and 7. K-Means was also

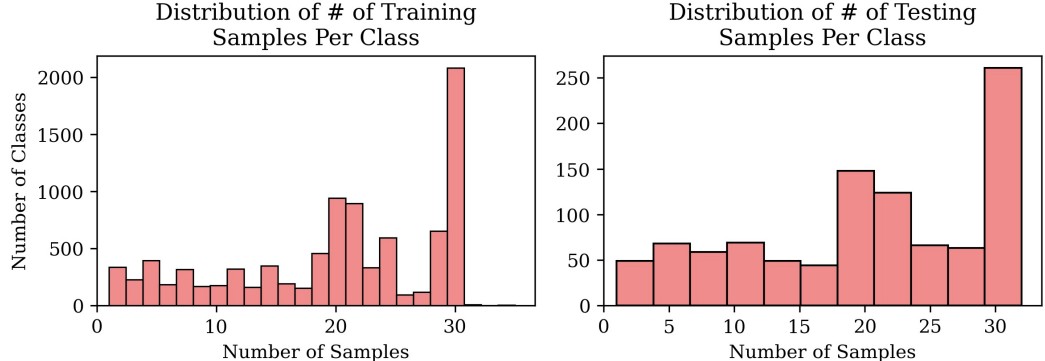

Figure 6: *Class distribution in CelebA.* Histograms visualizing the distribution over number of samples per class in the train (left) and test (right) datasets in CelebA.

|       | I    | II    | III   | IV    | V    | VI   |
|-------|------|-------|-------|-------|------|------|
| Train | 1.10 | 32.04 | 47.92 | 15.12 | 3.20 | 0.61 |
| Test  | 1.24 | 32.09 | 48.09 | 14.81 | 3.22 | 0.55 |

Table 4: *Summary statistics for CelebA Fitzpatrick Skintone.* The percentage of the dataset constituted by each Fitzpatrick Skintone in CelebA, in the train dataset and test dataset, respectively.

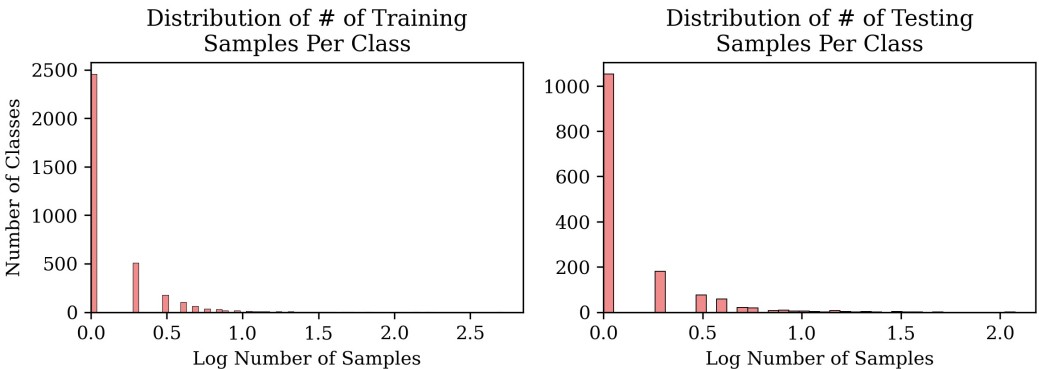

Figure 7: *Class distribution in LFW.* Histograms visualizing the distribution over logarithm of number of samples per class in the train (left) and test (right) datasets in LFW.

|       | Asian | Black | Indian | White |
|-------|-------|-------|--------|-------|
| Train | 8.43  | 4.17  | 1.71   | 85.70 |
| Test  | 6.27  | 4.61  | 1.79   | 87.33 |

Table 5: *Summary statistics for LFW Race* The percentage of the dataset constituted by each Race in LFW, in the train dataset and test dataset, respectively.

tested as a downstream classifier but showed poor performance. The impact of varying imbalance in the manually class imbalanced CARS196 experiments with all tested downstream classifiers is displayed in Table 8. Additional results for benchmarking of further fairness improvement methods in downstream classification of "imbalanced" embeddings (aside from naive use of balanced datasets) are shown in Table 8.

| | | | Overall | | | | |
| --- | --- | --- | --- | --- | --- | --- | --- | --- |
| | | | Contrastive · Distance | | Margin · Distance | | Margin · Semi-hard | |
| | | | Balanced | Imbalanced | Balanced | Imbalanced | Balanced | Imbalanced |
| UPSTREAM EMBEDDING | | Recall@1 | $0.861 \pm 0.003$ | $0.83 \pm 0.005$ | $0.854 \pm 0.002$ | $0.819 \pm 0.008$ | $0.83 \pm 0.002$ | $0.811 \pm 0.006$ |
| | | NMI | $0.909 \pm 0.003$ | $0.879 \pm 0.003$ | $0.894 \pm 0.003$ | $0.867 \pm 0.005$ | $0.876 \pm 0.004$ | $0.861 \pm 0.007$ |
| | | $U_{KL}$ | $0.433 \pm 0.004$ | $0.457 \pm 0.008$ | $0.096 \pm 0.002$ | $0.091 \pm 0.002$ | $0.133 \pm 0.004$ | $0.133 \pm 0.003$ |
| DOWNSTREAM CLASSIFICATION | LR | Accuracy | $0.878 \pm 0.002$ | $0.848 \pm 0.006$ | $0.88 \pm 0.002$ | $0.861 \pm 0.004$ | $0.858 \pm 0.005$ | $0.853 \pm 0.005$ |
| | | Precision | $0.877 \pm 0.002$ | $0.848 \pm 0.007$ | $0.883 \pm 0.002$ | $0.864 \pm 0.004$ | $0.86 \pm 0.005$ | $0.856 \pm 0.005$ |
| | | Recall | $0.876 \pm 0.002$ | $0.846 \pm 0.006$ | $0.879 \pm 0.002$ | $0.86 \pm 0.004$ | $0.856 \pm 0.005$ | $0.852 \pm 0.005$ |
| | RF | Accuracy | $0.855 \pm 0.002$ | $0.832 \pm 0.006$ | $0.816 \pm 0.003$ | $0.758 \pm 0.011$ | $0.819 \pm 0.005$ | $0.79 \pm 0.006$ |
| | | Precision | $0.859 \pm 0.003$ | $0.835 \pm 0.006$ | $0.82 \pm 0.003$ | $0.763 \pm 0.01$ | $0.822 \pm 0.005$ | $0.794 \pm 0.006$ |
| | | Recall | $0.855 \pm 0.003$ | $0.831 \pm 0.006$ | $0.815 \pm 0.003$ | $0.757 \pm 0.011$ | $0.817 \pm 0.005$ | $0.788 \pm 0.005$ |
| | SVM | Accuracy | $0.876 \pm 0.002$ | $0.852 \pm 0.006$ | $0.882 \pm 0.002$ | $0.863 \pm 0.004$ | $0.863 \pm 0.005$ | $0.86 \pm 0.004$ |
| | | Precision | $0.875 \pm 0.002$ | $0.855 \pm 0.007$ | $0.888 \pm 0.002$ | $0.875 \pm 0.003$ | $0.867 \pm 0.005$ | $0.867 \pm 0.004$ |
| | | Recall | $0.874 \pm 0.002$ | $0.85 \pm 0.006$ | $0.881 \pm 0.002$ | $0.863 \pm 0.004$ | $0.863 \pm 0.005$ | $0.86 \pm 0.004$ |

| | | | Overall | | | | | | | |
| --- | --- | --- | --- | --- | --- | --- | --- | --- | --- | --- |
| | | | Multisimilarity | | Proxy-NCA | | Triplet · Distance | | Triplet · Semi-hard | |
| | | | Balanced | Imbalanced | Balanced | Imbalanced | Balanced | Imbalanced | Balanced | Imbalanced |
| UPSTREAM EMBEDDING | | Recall@1 | $0.858 \pm 0.002$ | $0.839 \pm 0.005$ | $0.887 \pm 0.003$ | $0.858 \pm 0.005$ | $0.866 \pm 0.003$ | $0.848 \pm 0.006$ | $0.794 \pm 0.003$ | $0.778 \pm 0.006$ |
| | | NMI | $0.898 \pm 0.003$ | $0.881 \pm 0.004$ | $0.915 \pm 0.003$ | $0.89 \pm 0.005$ | $0.903 \pm 0.001$ | $0.884 \pm 0.005$ | $0.849 \pm 0.001$ | $0.834 \pm 0.007$ |
| | | $U_{KL}$ | $0.151 \pm 0.002$ | $0.155 \pm 0.003$ | $0.083 \pm 0.001$ | $0.094 \pm 0.003$ | $0.304 \pm 0.003$ | $0.293 \pm 0.003$ | $0.397 \pm 0.009$ | $0.382 \pm 0.007$ |
| DOWNSTREAM CLASSIFICATION | LR | Accuracy | $0.886 \pm 0.001$ | $0.875 \pm 0.005$ | $0.898 \pm 0.003$ | $0.877 \pm 0.005$ | $0.885 \pm 0.002$ | $0.872 \pm 0.004$ | $0.828 \pm 0.003$ | $0.818 \pm 0.005$ |
| | | Precision | $0.889 \pm 0.001$ | $0.878 \pm 0.004$ | $0.901 \pm 0.002$ | $0.879 \pm 0.005$ | $0.888 \pm 0.002$ | $0.874 \pm 0.003$ | $0.829 \pm 0.003$ | $0.821 \pm 0.005$ |
| | | Recall | $0.885 \pm 0.001$ | $0.873 \pm 0.005$ | $0.898 \pm 0.003$ | $0.876 \pm 0.005$ | $0.883 \pm 0.002$ | $0.87 \pm 0.004$ | $0.825 \pm 0.003$ | $0.816 \pm 0.005$ |
| | RF | Accuracy | $0.825 \pm 0.004$ | $0.794 \pm 0.005$ | $0.852 \pm 0.003$ | $0.823 \pm 0.007$ | $0.858 \pm 0.003$ | $0.836 \pm 0.004$ | $0.808 \pm 0.004$ | $0.792 \pm 0.008$ |
| | | Precision | $0.831 \pm 0.004$ | $0.797 \pm 0.006$ | $0.857 \pm 0.003$ | $0.825 \pm 0.008$ | $0.861 \pm 0.003$ | $0.838 \pm 0.004$ | $0.811 \pm 0.004$ | $0.795 \pm 0.008$ |
| | | Recall | $0.824 \pm 0.004$ | $0.793 \pm 0.005$ | $0.852 \pm 0.003$ | $0.823 \pm 0.008$ | $0.857 \pm 0.003$ | $0.835 \pm 0.004$ | $0.807 \pm 0.004$ | $0.791 \pm 0.008$ |
| | SVM | Accuracy | $0.888 \pm 0.001$ | $0.876 \pm 0.004$ | $0.894 \pm 0.002$ | $0.871 \pm 0.003$ | $0.887 \pm 0.002$ | $0.878 \pm 0.004$ | $0.835 \pm 0.003$ | $0.83 \pm 0.005$ |
| | | Precision | $0.893 \pm 0.001$ | $0.886 \pm 0.004$ | $0.902 \pm 0.001$ | $0.887 \pm 0.003$ | $0.892 \pm 0.002$ | $0.884 \pm 0.004$ | $0.839 \pm 0.003$ | $0.834 \pm 0.005$ |
| | | Recall | $0.887 \pm 0.001$ | $0.876 \pm 0.004$ | $0.894 \pm 0.002$ | $0.871 \pm 0.003$ | $0.886 \pm 0.003$ | $0.877 \pm 0.003$ | $0.834 \pm 0.003$ | $0.829 \pm 0.005$ |

Table 6: *Overall results on CARS196.* Metrics over entire test dataset in representation space and downstream classification (LR, RF, and SVM) over 10 seed in manually class imbalanced experiments (Imbalanced) and control experiments (Balanced) for CARS196.

| | | | Subgroup Gap | | | | | |
| --- | --- | --- | --- | --- | --- | --- | --- | --- |
| | | | Contrastive · Distance | | Margin · Distance | | Margin · Semi-hard | |
| | | | Balanced | Imbalanced | Balanced | Imbalanced | Balanced | Imbalanced |
| UPSTREAM EMBEDDING | | Recall@1 | $0.861 \pm 0.003$ | $0.83 \pm 0.005$ | $0.854 \pm 0.002$ | $0.819 \pm 0.008$ | $0.83 \pm 0.002$ | |
| | | NMI | $-0.013 \pm 0.004$ | $0.106 \pm 0.013$ | $-0.016 \pm 0.005$ | $0.124 \pm 0.014$ | $-0.018 \pm 0.006$ | $0.11 \pm 0.016$ |
| | | $U_{KL}$ | $-0.093 \pm 0.004$ | $0.011 \pm 0.011$ | $-0.033 \pm 0.003$ | $0.01 \pm 0.003$ | $-0.038 \pm 0.006$ | $0.012 \pm 0.005$ |
| DOWNSTREAM CLASSIFICATION | LR | Accuracy | $0.002 \pm 0.003$ | $0.147 \pm 0.023$ | $0.003 \pm 0.003$ | $0.12 \pm 0.013$ | $0.004 \pm 0.007$ | $0.115 \pm 0.018$ |
| | | Precision | $0.336 \pm 0.004$ | $0.407 \pm 0.013$ | $0.355 \pm 0.008$ | $0.402 \pm 0.013$ | $0.353 \pm 0.008$ | $0.403 \pm 0.013$ |
| | | Recall | $0.351 \pm 0.004$ | $0.439 \pm 0.016$ | $0.368 \pm 0.008$ | $0.426 \pm 0.015$ | $0.367 \pm 0.008$ | $0.431 \pm 0.014$ |
| | RF | Accuracy | $0.001 \pm 0.006$ | $0.115 \pm 0.021$ | $0.002 \pm 0.004$ | $0.315 \pm 0.022$ | $0.002 \pm 0.008$ | $0.231 \pm 0.024$ |
| | | Precision | $0.358 \pm 0.005$ | $0.396 \pm 0.013$ | $0.374 \pm 0.006$ | $0.441 \pm 0.013$ | $0.362 \pm 0.007$ | $0.429 \pm 0.014$ |
| | | Recall | $0.373 \pm 0.005$ | $0.409 \pm 0.015$ | $0.387 \pm 0.006$ | $0.502 \pm 0.013$ | $0.376 \pm 0.008$ | $0.481 \pm 0.016$ |
| | SVM | Accuracy | $0.003 \pm 0.004$ | $0.086 \pm 0.022$ | $0.002 \pm 0.003$ | $0.039 \pm 0.013$ | $0.002 \pm 0.006$ | $0.055 \pm 0.018$ |
| | | Precision | $0.33 \pm 0.006$ | $0.332 \pm 0.023$ | $0.35 \pm 0.008$ | $0.283 \pm 0.024$ | $0.347 \pm 0.011$ | $0.328 \pm 0.018$ |
| | | Recall | $0.347 \pm 0.004$ | $0.338 \pm 0.027$ | $0.363 \pm 0.008$ | $0.27 \pm 0.027$ | $0.361 \pm 0.011$ | $0.327 \pm 0.018$ |

| | | | Subgroup Gap | | | | | | | |
| --- | --- | --- | --- | --- | --- | --- | --- | --- | --- | --- |
| | | | Multisimilarity | | Proxy-NCA | | Triplet · Distance | | Triplet · Semi-hard | |
| | | | Balanced | Imbalanced | Balanced | Imbalanced | Balanced | Imbalanced | Balanced | Imbalanced |
| UPSTREAM EMBEDDING | | Recall@1 | $0.858 \pm 0.002$ | $0.839 \pm 0.005$ | $0.887 \pm 0.001$ | $0.858 \pm 0.005$ | $0.866 \pm 0.003$ | $0.848 \pm 0.006$ | $0.794 \pm 0.003$ | $0.778 \pm 0.006$ |
| | | NMI | $-0.014 \pm 0.004$ | $0.116 \pm 0.014$ | $-0.011 \pm 0.004$ | $0.148 \pm 0.014$ | $-0.013 \pm 0.001$ | $0.109 \pm 0.016$ | $-0.018 \pm 0.002$ | $0.083 \pm 0.015$ |
| | | $U_{KL}$ | $-0.03 \pm 0.003$ | $0.005 \pm 0.004$ | $-0.129 \pm 0.002$ | $0.005 \pm 0.005$ | $-0.047 \pm 0.005$ | $0.011 \pm 0.006$ | $-0.034 \pm 0.013$ | $0.023 \pm 0.011$ |
| DOWNSTREAM CLASSIFICATION | LR | Accuracy | $0.002 \pm 0.002$ | $0.117 \pm 0.018$ | $0.003 \pm 0.005$ | $0.155 \pm 0.019$ | $0.001 \pm 0.004$ | $0.131 \pm 0.017$ | $0.004 \pm 0.005$ | $0.12 \pm 0.016$ |
| | | Precision | $0.352 \pm 0.006$ | $0.396 \pm 0.014$ | $0.334 \pm 0.006$ | $0.438 \pm 0.012$ | $0.343 \pm 0.005$ | $0.404 \pm 0.011$ | $0.357 \pm 0.004$ | $0.392 \pm 0.009$ |
| | | Recall | $0.365 \pm 0.006$ | $0.418 \pm 0.018$ | $0.347 \pm 0.006$ | $0.46 \pm 0.016$ | $0.356 \pm 0.005$ | $0.431 \pm 0.013$ | $0.371 \pm 0.004$ | $0.421 \pm 0.01$ |
| | RF | Accuracy | $0.0 \pm 0.006$ | $0.26 \pm 0.023$ | $0.0 \pm 0.005$ | $0.221 \pm 0.022$ | $0.0 \pm 0.004$ | $0.168 \pm 0.015$ | $0.003 \pm 0.005$ | $0.141 \pm 0.02$ |
| | | Precision | $0.378 \pm 0.005$ | $0.441 \pm 0.011$ | $0.379 \pm 0.006$ | $0.458 \pm 0.01$ | $0.361 \pm 0.006$ | $0.425 \pm 0.008$ | $0.361 \pm 0.004$ | $0.393 \pm 0.012$ |
| | | Recall | $0.389 \pm 0.005$ | $0.485 \pm 0.009$ | $0.39 \pm 0.005$ | $0.476 \pm 0.011$ | $0.375 \pm 0.006$ | $0.45 \pm 0.009$ | $0.374 \pm 0.004$ | $0.426 \pm 0.012$ |
| | SVM | Accuracy | $0.002 \pm 0.002$ | $0.047 \pm 0.015$ | $0.0 \pm 0.003$ | $0.076 \pm 0.026$ | $0.0 \pm 0.004$ | $0.063 \pm 0.018$ | $0.002 \pm 0.005$ | $0.073 \pm 0.02$ |
| | | Precision | $0.349 \pm 0.006$ | $0.267 \pm 0.03$ | $0.335 \pm 0.006$ | $0.29 \pm 0.026$ | $0.339 \pm 0.007$ | $0.297 \pm 0.03$ | $0.354 \pm 0.007$ | $0.364 \pm 0.015$ |
| | | Recall | $0.362 \pm 0.005$ | $0.258 \pm 0.032$ | $0.349 \pm 0.006$ | $0.273 \pm 0.029$ | $0.353 \pm 0.007$ | $0.295 \pm 0.03$ | $0.368 \pm 0.006$ | $0.372 \pm 0.016$ |

Table 7: *Gap study on CARS196.* Average gaps in representation space and downstream classification (LR, RF, and SVM) over 10 seeds between minoritized and majoritized classes in manually class imbalanced experiments (Imbalanced) and control experiments (Balanced) for CARS196.

## C.2   CUB200

Additional results for all loss and batch mining strategies for the manually class imbalanced experiments and balanced controls for CUB200 are located in Tables 9 and 10. K-Means was also tested as a downstream classifier but showed poor performance. The impact of varying imbalance in the manually class imbalanced experiments in the upstream embedding, and all tested downstream classifiers is displayed in Table 9. Additional results for benchmarking of further fairness improvement methods

Table 8: *Benchmarking additional fairness improvement methods in downstream classification on CARS196 (Classes).* Overall performance and subgroup gaps for Domain-Independent Training and Oversampling (Wang et al., 2020b) on CARS196 in class imbalanced experiments with upstream embedding trained on imbalanced dataset.

(a) Domain-Independent Training

| METRIC ↓ | Contr. (D) | Margin (D) | Margin (Sem.) | Msim. | ProxyNCA | Triplet (D) | Triplet (S) |
|---|---|---|---|---|---|---|---|
| **Overall** | | | | | | | |
| ACCURACY | $0.812 \pm 0.011$ | $0.834 \pm 0.009$ | $0.820 \pm 0.008$ | $0.842 \pm 0.008$ | $0.869 \pm 0.005$ | $0.836 \pm 0.007$ | $0.742 \pm 0.007$ |
| PRECISION | $0.834 \pm 0.009$ | $0.861 \pm 0.004$ | $0.847 \pm 0.004$ | $0.872 \pm 0.004$ | $0.878 \pm 0.005$ | $0.865 \pm 0.009$ | $0.804 \pm 0.008$ |
| RECALL | $0.811 \pm 0.011$ | $0.833 \pm 0.009$ | $0.818 \pm 0.008$ | $0.840 \pm 0.008$ | $0.869 \pm 0.005$ | $0.834 \pm 0.008$ | $0.740 \pm 0.008$ |
| **Gap** | | | | | | | |
| ACCURACY | $0.001 \pm 0.027$ | $0.010 \pm 0.018$ | $0.017 \pm 0.021$ | $0.018 \pm 0.018$ | $0.120 \pm 0.018$ | $0.022 \pm 0.017$ | $0.094 \pm 0.021$ |
| PRECISION | $0.304 \pm 0.021$ | $0.275 \pm 0.021$ | $0.313 \pm 0.019$ | $0.247 \pm 0.027$ | $0.398 \pm 0.015$ | $0.236 \pm 0.022$ | $0.289 \pm 0.016$ |
| RECALL | $0.260 \pm 0.024$ | $0.218 \pm 0.022$ | $0.258 \pm 0.018$ | $0.182 \pm 0.027$ | $0.398 \pm 0.018$ | $0.175 \pm 0.022$ | $0.177 \pm 0.016$ |

(b) Oversampling

| METRIC ↓ | Contr. (D) | Margin (D) | Margin (Sem.) | Msim. | ProxyNCA | Triplet (D) | Triplet (S) |
|---|---|---|---|---|---|---|---|
| **Overall** | | | | | | | |
| ACCURACY | $0.851 \pm 0.007$ | $0.862 \pm 0.004$ | $0.853 \pm 0.006$ | $0.875 \pm 0.004$ | $0.878 \pm 0.004$ | $0.875 \pm 0.005$ | $0.820 \pm 0.005$ |
| PRECISION | $0.854 \pm 0.006$ | $0.864 \pm 0.004$ | $0.855 \pm 0.006$ | $0.877 \pm 0.004$ | $0.880 \pm 0.005$ | $0.876 \pm 0.004$ | $0.822 \pm 0.005$ |
| RECALL | $0.853 \pm 0.006$ | $0.862 \pm 0.004$ | $0.853 \pm 0.006$ | $0.875 \pm 0.004$ | $0.878 \pm 0.004$ | $0.875 \pm 0.004$ | $0.821 \pm 0.005$ |
| **Gap** | | | | | | | |
| ACCURACY | $0.128 \pm 0.023$ | $0.099 \pm 0.014$ | $0.102 \pm 0.020$ | $0.097 \pm 0.019$ | $0.136 \pm 0.017$ | $0.108 \pm 0.018$ | $0.109 \pm 0.019$ |
| PRECISION | $0.398 \pm 0.012$ | $0.386 \pm 0.017$ | $0.391 \pm 0.014$ | $0.383 \pm 0.015$ | $0.422 \pm 0.014$ | $0.387 \pm 0.013$ | $0.387 \pm 0.011$ |
| RECALL | $0.423 \pm 0.015$ | $0.403 \pm 0.017$ | $0.414 \pm 0.014$ | $0.396 \pm 0.019$ | $0.436 \pm 0.017$ | $0.406 \pm 0.015$ | $0.413 \pm 0.012$ |

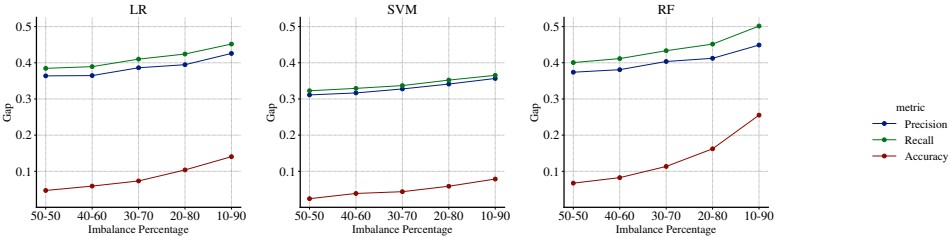

Figure 8: Impact of varying imbalance between the *minoritized* and *majoritized* classes on various downstream classifiers (RF, LR and SVM) in the manually class imbalanced CARS196 experiments. (Note: the imbalance percentage $50 - 50$ is equivalent to the balanced setting). Gaps increase for all classifiers downstream with more imbalance introduced to the upstream training data.

in downstream classification of "imbalanced" embeddings (aside from naive use of balanced datasets) are shown in Table 11. Benchmarking of fairness improvement methods in downstream classification for bird color are shown in Table 12. Per-subgroup and overall results for CUB200 color experiments with standard margin-distance and PARADE are displayed in Table 13.

## C.3 CELEBA

Additional results for all loss and batch mining strategies for the CelebA dataset are located in Tables 14 and 15. Additional PARADE results for subgroup gaps excluding Fitzpatrick Skintone VI (as mentioned in Section 6.3) are located in Table 16.

| | | | Overall | | | | | |
| | | | Contrastive · Distance | | Margin · Distance | | Margin · Semi-hard | |
| | | | Balanced | Imbalanced | Balanced | Imbalanced | Balanced | Imbalanced |
|---|---|---|---|---|---|---|---|---|
| UPSTREAM EMBEDDING | | Recall@1 | $0.79 \pm 0.002$ | $0.782 \pm 0.005$ | $0.786 \pm 0.003$ | $0.78 \pm 0.005$ | $0.775 \pm 0.006$ | $0.766 \pm 0.006$ |
| | | NMI | $0.872 \pm 0.002$ | $0.859 \pm 0.003$ | $0.861 \pm 0.003$ | $0.856 \pm 0.004$ | $0.856 \pm 0.003$ | $0.85 \pm 0.005$ |
| | | $U_{KL}$ | $0.397 \pm 0.003$ | $0.449 \pm 0.007$ | $0.076 \pm 0.001$ | $0.076 \pm 0.001$ | $0.113 \pm 0.003$ | $0.113 \pm 0.002$ |
| DOWNSTREAM CLASSIFICATION | LR | Accuracy | $0.815 \pm 0.002$ | $0.81 \pm 0.005$ | $0.815 \pm 0.002$ | $0.827 \pm 0.006$ | $0.809 \pm 0.004$ | $0.818 \pm 0.005$ |
| | | Precision | $0.817 \pm 0.002$ | $0.811 \pm 0.005$ | $0.822 \pm 0.001$ | $0.831 \pm 0.006$ | $0.815 \pm 0.004$ | $0.822 \pm 0.005$ |
| | | Recall | $0.815 \pm 0.002$ | $0.81 \pm 0.005$ | $0.816 \pm 0.002$ | $0.827 \pm 0.006$ | $0.81 \pm 0.004$ | $0.818 \pm 0.005$ |
| | RF | Accuracy | $0.776 \pm 0.002$ | $0.787 \pm 0.007$ | $0.757 \pm 0.003$ | $0.735 \pm 0.009$ | $0.768 \pm 0.002$ | $0.758 \pm 0.004$ |
| | | Precision | $0.785 \pm 0.003$ | $0.793 \pm 0.006$ | $0.762 \pm 0.003$ | $0.737 \pm 0.01$ | $0.774 \pm 0.002$ | $0.762 \pm 0.004$ |
| | | Recall | $0.776 \pm 0.002$ | $0.787 \pm 0.007$ | $0.757 \pm 0.003$ | $0.735 \pm 0.009$ | $0.769 \pm 0.002$ | $0.758 \pm 0.004$ |
| | SVM | Accuracy | $0.813 \pm 0.002$ | $0.811 \pm 0.007$ | $0.81 \pm 0.002$ | $0.82 \pm 0.007$ | $0.808 \pm 0.004$ | $0.815 \pm 0.004$ |
| | | Precision | $0.823 \pm 0.003$ | $0.821 \pm 0.007$ | $0.827 \pm 0.002$ | $0.843 \pm 0.005$ | $0.818 \pm 0.003$ | $0.829 \pm 0.004$ |
| | | Recall | $0.813 \pm 0.002$ | $0.811 \pm 0.007$ | $0.811 \pm 0.002$ | $0.82 \pm 0.006$ | $0.808 \pm 0.003$ | $0.815 \pm 0.004$ |

| | | | Overall | | | | | | | |
| | | | Multisimilarity | | Proxy-NCA | | Triplet · Distance | | Triplet · Semi-hard | |
| | | | Balanced | Imbalanced | Balanced | Imbalanced | Balanced | Imbalanced | Balanced | Imbalanced |
|---|---|---|---|---|---|---|---|---|---|---|
| UPSTREAM EMBEDDING | | Recall@1 | $0.779 \pm 0.006$ | $0.788 \pm 0.005$ | $0.807 \pm 0.004$ | $0.8 \pm 0.007$ | $0.792 \pm 0.003$ | $0.795 \pm 0.007$ | $0.761 \pm 0.004$ | |
| | | NMI | $0.857 \pm 0.003$ | $0.857 \pm 0.004$ | $0.873 \pm 0.003$ | $0.86 \pm 0.005$ | $0.866 \pm 0.002$ | $0.861 \pm 0.005$ | $0.848 \pm 0.005$ | $0.843 \pm 0.004$ |
| | | $U_{KL}$ | $0.139 \pm 0.001$ | $0.146 \pm 0.001$ | $0.056 \pm 0.001$ | $0.073 \pm 0.001$ | $0.274 \pm 0.005$ | $0.277 \pm 0.005$ | $0.336 \pm 0.004$ | $0.321 \pm 0.004$ |
| DOWNSTREAM CLASSIFICATION | LR | Accuracy | $0.813 \pm 0.004$ | $0.833 \pm 0.004$ | $0.824 \pm 0.003$ | $0.828 \pm 0.005$ | $0.82 \pm 0.001$ | $0.828 \pm 0.005$ | $0.802 \pm 0.003$ | $0.806 \pm 0.004$ |
| | | Precision | $0.82 \pm 0.004$ | $0.836 \pm 0.004$ | $0.828 \pm 0.003$ | $0.833 \pm 0.005$ | $0.826 \pm 0.001$ | $0.833 \pm 0.006$ | $0.808 \pm 0.003$ | $0.811 \pm 0.004$ |
| | | Recall | $0.814 \pm 0.004$ | $0.833 \pm 0.004$ | $0.824 \pm 0.003$ | $0.828 \pm 0.005$ | $0.82 \pm 0.001$ | $0.828 \pm 0.005$ | $0.803 \pm 0.003$ | $0.806 \pm 0.005$ |
| | RF | Accuracy | $0.754 \pm 0.003$ | $0.754 \pm 0.005$ | $0.761 \pm 0.006$ | $0.768 \pm 0.007$ | $0.786 \pm 0.004$ | $0.789 \pm 0.008$ | $0.776 \pm 0.003$ | $0.774 \pm 0.007$ |
| | | Precision | $0.76 \pm 0.004$ | $0.755 \pm 0.005$ | $0.768 \pm 0.007$ | $0.774 \pm 0.006$ | $0.794 \pm 0.004$ | $0.792 \pm 0.009$ | $0.782 \pm 0.003$ | $0.778 \pm 0.006$ |
| | | Recall | $0.754 \pm 0.003$ | $0.755 \pm 0.005$ | $0.762 \pm 0.006$ | $0.768 \pm 0.007$ | $0.786 \pm 0.004$ | $0.789 \pm 0.008$ | $0.777 \pm 0.003$ | $0.774 \pm 0.007$ |
| | SVM | Accuracy | $0.812 \pm 0.002$ | $0.828 \pm 0.006$ | $0.818 \pm 0.002$ | $0.816 \pm 0.005$ | $0.819 \pm 0.002$ | $0.83 \pm 0.005$ | $0.798 \pm 0.001$ | $0.808 \pm 0.005$ |
| | | Precision | $0.825 \pm 0.003$ | $0.848 \pm 0.005$ | $0.834 \pm 0.002$ | $0.852 \pm 0.004$ | $0.829 \pm 0.003$ | $0.845 \pm 0.004$ | $0.806 \pm 0.002$ | $0.816 \pm 0.005$ |
| | | Recall | $0.812 \pm 0.002$ | $0.828 \pm 0.006$ | $0.819 \pm 0.002$ | $0.817 \pm 0.005$ | $0.819 \pm 0.002$ | $0.831 \pm 0.005$ | $0.798 \pm 0.001$ | $0.808 \pm 0.005$ |

Table 9: *Overall results on CUB200.* Metrics over entire test dataset in representation space and downstream classification (LR, RF, and SVM) over 10 seed in manually class imbalanced experiments (Imbalanced) and control experiments (Balanced) for CUB200.

| | | | Subgroup Gap | | | | | |
| | | | Contrastive · Distance | | Margin · Distance | | Margin · Semi-hard | |
| | | | Balanced | Imbalanced | Balanced | Imbalanced | Balanced | Imbalanced |
|---|---|---|---|---|---|---|---|---|
| UPSTREAM EMBEDDING | | Recall@1 | $0.011 \pm 0.004$ | $0.168 \pm 0.028$ | $0.008 \pm 0.005$ | $0.212 \pm 0.029$ | $0.01 \pm 0.008$ | $0.187 \pm 0.031$ |
| | | NMI | $-0.009 \pm 0.002$ | $0.109 \pm 0.015$ | $-0.008 \pm 0.005$ | $0.112 \pm 0.012$ | $-0.009 \pm 0.003$ | $0.092 \pm 0.017$ |
| | | $U_{KL}$ | $-0.112 \pm 0.004$ | $0.004 \pm 0.011$ | $-0.043 \pm 0.002$ | $0.0 \pm 0.002$ | $-0.05 \pm 0.004$ | $0.002 \pm 0.004$ |
| DOWNSTREAM CLASSIFICATION | LR | Accuracy | $0.014 \pm 0.004$ | $0.181 \pm 0.029$ | $0.008 \pm 0.003$ | $0.131 \pm 0.027$ | $0.009 \pm 0.006$ | $0.131 \pm 0.031$ |
| | | Precision | $0.333 \pm 0.003$ | $0.417 \pm 0.012$ | $0.337 \pm 0.005$ | $0.39 \pm 0.014$ | $0.331 \pm 0.006$ | $0.393 \pm 0.015$ |
| | | Recall | $0.354 \pm 0.004$ | $0.462 \pm 0.016$ | $0.356 \pm 0.005$ | $0.424 \pm 0.018$ | $0.351 \pm 0.007$ | $0.43 \pm 0.016$ |
| | RF | Accuracy | $0.013 \pm 0.004$ | $0.121 \pm 0.026$ | $0.009 \pm 0.005$ | $0.325 \pm 0.035$ | $0.01 \pm 0.006$ | $0.255 \pm 0.031$ |
| | | Precision | $0.339 \pm 0.007$ | $0.386 \pm 0.014$ | $0.347 \pm 0.006$ | $0.428 \pm 0.011$ | $0.342 \pm 0.007$ | $0.418 \pm 0.014$ |
| | | Recall | $0.359 \pm 0.006$ | $0.391 \pm 0.015$ | $0.365 \pm 0.006$ | $0.495 \pm 0.01$ | $0.362 \pm 0.007$ | $0.478 \pm 0.014$ |
| | SVM | Accuracy | $0.014 \pm 0.003$ | $0.106 \pm 0.032$ | $0.009 \pm 0.004$ | $0.043 \pm 0.028$ | $0.009 \pm 0.006$ | $0.058 \pm 0.029$ |
| | | Precision | $0.326 \pm 0.008$ | $0.36 \pm 0.021$ | $0.332 \pm 0.008$ | $0.301 \pm 0.023$ | $0.329 \pm 0.006$ | $0.329 \pm 0.017$ |
| | | Recall | $0.345 \pm 0.007$ | $0.362 \pm 0.024$ | $0.348 \pm 0.008$ | $0.278 \pm 0.027$ | $0.348 \pm 0.006$ | $0.323 \pm 0.02$ |

| | | | Subgroup Gap | | | | | | | |
| | | | Multisimilarity | | Proxy-NCA | | Triplet · Distance | | Triplet · Semi-hard | |
| | | | Balanced | Imbalanced | Balanced | Imbalanced | Balanced | Imbalanced | Balanced | Imbalanced |
|---|---|---|---|---|---|---|---|---|---|---|
| UPSTREAM EMBEDDING | | Recall@1 | $0.008 \pm 0.009$ | $0.187 \pm 0.031$ | $0.01 \pm 0.005$ | $0.256 \pm 0.03$ | $0.009 \pm 0.004$ | $0.159 \pm 0.031$ | $0.009 \pm 0.006$ | $0.168 \pm 0.036$ |
| | | NMI | $-0.008 \pm 0.004$ | $0.113 \pm 0.016$ | $-0.009 \pm 0.004$ | $0.142 \pm 0.015$ | $-0.007 \pm 0.003$ | $0.103 \pm 0.016$ | $-0.01 \pm 0.006$ | $0.082 \pm 0.016$ |
| | | $U_{KL}$ | $-0.036 \pm 0.002$ | $-0.003 \pm 0.003$ | $-0.131 \pm 0.003$ | $-0.012 \pm 0.005$ | $-0.057 \pm 0.006$ | $-0.004 \pm 0.009$ | $-0.051 \pm 0.005$ | $0.014 \pm 0.011$ |
| DOWNSTREAM CLASSIFICATION | LR | Accuracy | $0.009 \pm 0.006$ | $0.141 \pm 0.032$ | $0.007 \pm 0.005$ | $0.169 \pm 0.027$ | $0.011 \pm 0.002$ | $0.179 \pm 0.031$ | $0.011 \pm 0.005$ | $0.134 \pm 0.031$ |
| | | Precision | $0.337 \pm 0.008$ | $0.391 \pm 0.016$ | $0.337 \pm 0.005$ | $0.428 \pm 0.018$ | $0.335 \pm 0.005$ | $0.41 \pm 0.014$ | $0.336 \pm 0.006$ | $0.384 \pm 0.014$ |
| | | Recall | $0.356 \pm 0.008$ | $0.427 \pm 0.019$ | $0.356 \pm 0.006$ | $0.455 \pm 0.019$ | $0.355 \pm 0.004$ | $0.459 \pm 0.016$ | $0.357 \pm 0.006$ | $0.426 \pm 0.016$ |
| | RF | Accuracy | $0.009 \pm 0.006$ | $0.282 \pm 0.034$ | $0.009 \pm 0.009$ | $0.214 \pm 0.026$ | $0.01 \pm 0.005$ | $0.192 \pm 0.035$ | $0.011 \pm 0.004$ | $0.175 \pm 0.03$ |
| | | Precision | $0.348 \pm 0.006$ | $0.428 \pm 0.011$ | $0.355 \pm 0.01$ | $0.436 \pm 0.009$ | $0.347 \pm 0.006$ | $0.409 \pm 0.014$ | $0.341 \pm 0.007$ | $0.393 \pm 0.013$ |
| | | Recall | $0.365 \pm 0.005$ | $0.48 \pm 0.012$ | $0.372 \pm 0.01$ | $0.443 \pm 0.01$ | $0.364 \pm 0.007$ | $0.44 \pm 0.014$ | $0.36 \pm 0.005$ | $0.437 \pm 0.015$ |
| | SVM | Accuracy | $0.009 \pm 0.003$ | $0.048 \pm 0.027$ | $0.009 \pm 0.003$ | $0.063 \pm 0.026$ | $0.011 \pm 0.003$ | $0.071 \pm 0.027$ | $0.012 \pm 0.002$ | $0.082 \pm 0.03$ |
| | | Precision | $0.335 \pm 0.005$ | $0.307 \pm 0.02$ | $0.33 \pm 0.006$ | $0.347 \pm 0.022$ | $0.334 \pm 0.006$ | $0.324 \pm 0.014$ | $0.334 \pm 0.005$ | $0.34 \pm 0.012$ |
| | | Recall | $0.352 \pm 0.004$ | $0.284 \pm 0.023$ | $0.347 \pm 0.004$ | $0.299 \pm 0.022$ | $0.353 \pm 0.006$ | $0.315 \pm 0.017$ | $0.355 \pm 0.005$ | $0.356 \pm 0.014$ |

Table 10: *Gap study on CUB200.* Average gaps in representation space and downstream classification (LR, RF, and SVM) over 10 seeds between minoritized and majoritized classes in manually class imbalanced experiments (Imbalanced) and control experiments (Balanced) for CUB200.

## C.4 LFW

Additional results for all loss and batch mining strategies for the LFW dataset are located in Tables 18 and 19. Per-subgroup results for LFW to demonstrate worst-group performance for the "White" subgroup (as mentioned in Section 6.3) are located in Table 20. Benchmarking of fairness improvement methods in downstream classification for bird color are shown in Table 21.

Table 11: *Benchmarking additional fairness improvement methods in downstream classification on CUB200 (Classes).* Overall performance and subgroup gaps for Domain-Independent Training and Oversampling (Wang et al., 2020b) on CUB200-2011 in class imbalanced experiments with upstream embedding trained on imbalanced dataset.

(a) Domain-Independent Training

| METRIC ↓ | Contr. (D) | Margin (D) | Margin (Sem.) | Msim. | ProxyNCA | Triplet (D) | Triplet (S) |
|---|---|---|---|---|---|---|---|
| **Overall** | | | | | | | |
| ACCURACY | $0.782 \pm 0.008$ | $0.809 \pm 0.004$ | $0.794 \pm 0.005$ | $0.805 \pm 0.004$ | $0.823 \pm 0.006$ | $0.798 \pm 0.005$ | $0.749 \pm 0.006$ |
| PRECISION | $0.805 \pm 0.011$ | $0.840 \pm 0.006$ | $0.827 \pm 0.005$ | $0.847 \pm 0.005$ | $0.836 \pm 0.006$ | $0.842 \pm 0.005$ | $0.806 \pm 0.006$ |
| RECALL | $0.782 \pm 0.008$ | $0.809 \pm 0.004$ | $0.795 \pm 0.005$ | $0.805 \pm 0.004$ | $0.823 \pm 0.006$ | $0.798 \pm 0.004$ | $0.749 \pm 0.005$ |
| **Gap** | | | | | | | |
| ACCURACY | $0.034 \pm 0.033$ | $0.003 \pm 0.030$ | $0.014 \pm 0.031$ | $0.024 \pm 0.031$ | $0.140 \pm 0.030$ | $0.024 \pm 0.030$ | $0.077 \pm 0.031$ |
| PRECISION | $0.340 \pm 0.024$ | $0.304 \pm 0.031$ | $0.301 \pm 0.022$ | $0.264 \pm 0.028$ | $0.410 \pm 0.022$ | $0.262 \pm 0.032$ | $0.270 \pm 0.018$ |
| RECALL | $0.308 \pm 0.027$ | $0.253 \pm 0.035$ | $0.249 \pm 0.025$ | $0.189 \pm 0.031$ | $0.415 \pm 0.024$ | $0.188 \pm 0.036$ | $0.177 \pm 0.021$ |

(b) Oversampling

| METRIC ↓ | Contr. (D) | Margin (D) | Margin (Sem.) | Msim. | ProxyNCA | Triplet (D) | Triplet (S) |
|---|---|---|---|---|---|---|---|
| **Overall** | | | | | | | |
| ACCURACY | $0.811 \pm 0.005$ | $0.828 \pm 0.005$ | $0.818 \pm 0.004$ | $0.832 \pm 0.005$ | $0.828 \pm 0.005$ | $0.829 \pm 0.006$ | $0.806 \pm 0.005$ |
| PRECISION | $0.814 \pm 0.005$ | $0.831 \pm 0.005$ | $0.822 \pm 0.004$ | $0.835 \pm 0.005$ | $0.833 \pm 0.005$ | $0.832 \pm 0.006$ | $0.811 \pm 0.004$ |
| RECALL | $0.812 \pm 0.005$ | $0.828 \pm 0.005$ | $0.819 \pm 0.004$ | $0.833 \pm 0.005$ | $0.828 \pm 0.005$ | $0.829 \pm 0.006$ | $0.807 \pm 0.005$ |
| **Gap** | | | | | | | |
| ACCURACY | $0.182 \pm 0.027$ | $0.131 \pm 0.028$ | $0.129 \pm 0.030$ | $0.142 \pm 0.035$ | $0.170 \pm 0.026$ | $0.177 \pm 0.032$ | $0.135 \pm 0.032$ |
| PRECISION | $0.421 \pm 0.011$ | $0.386 \pm 0.015$ | $0.391 \pm 0.016$ | $0.391 \pm 0.016$ | $0.428 \pm 0.016$ | $0.409 \pm 0.015$ | $0.385 \pm 0.013$ |
| RECALL | $0.464 \pm 0.015$ | $0.420 \pm 0.018$ | $0.428 \pm 0.018$ | $0.426 \pm 0.020$ | $0.455 \pm 0.017$ | $0.457 \pm 0.017$ | $0.427 \pm 0.015$ |

Table 12: *Benchmarking additional fairness improvement methods in downstream classification on CUB200 (Color).* Overall performance and subgroup gaps for Domain-Independent Training and Oversampling (Wang et al., 2020b) on CUB200-2011 in bird color experiments.

(a) Domain-Independent Training

| METRIC ↓ | Margin (D) |
|---|---|
| **Overall** | |
| ACCURACY | $0.490 \pm 0.005$ |
| PRECISION | $0.896 \pm 0.003$ |
| RECALL | $0.489 \pm 0.006$ |
| **Gap** | |
| ACCURACY | $0.426 \pm 0.017$ |
| PRECISION | $0.185 \pm 0.108$ |
| RECALL | $0.353 \pm 0.108$ |

(b) Oversampling

| METRIC ↓ | Margin (D) |
|---|---|
| **Overall** | |
| ACCURACY | $0.802 \pm 0.002$ |
| PRECISION | $0.816 \pm 0.002$ |
| RECALL | $0.802 \pm 0.002$ |
| **Gap** | |
| ACCURACY | $0.143 \pm 0.019$ |
| PRECISION | $0.323 \pm 0.063$ |
| RECALL | $0.348 \pm 0.064$ |

## C.5 EXPLORATION OF FAIRNESS - UTILITY TRADEOFF AND VARYING HYPERPARAMETERS IN PARADE

We vary $\alpha_{SA}$ and $\rho$ in the PARADE objective to explore the relationship between the overall performance, subgroup gap, and worst-group performance in PARADE. As stated in the main paper, we optimize $\alpha_{SA}$ and $\rho$ via worst-group performance. Results of this analysis are displayed in Figure 12. We use our exploration to expound on how to optimize for $\alpha_{SA}$ and $\rho$. As seen in Figure 12, a clear trend that inversely relates overall performance, and fairness as measured by subgroup gap and worst-group performance is seen for the uniformity metric, $U_{KL}$ over the grid of $\alpha_{SA}$ and $\rho$ values (Note that higher values of $U_{KL}$ correspond to *worse* performance). Recall@1 and NMI demonstrate noisier relationships between overall performance and fairness; and several $\alpha_{SA}$, $\rho$ choices appear to select an optimal tradeoff. In Figure 12, for Recall@1, we observe that at the location $\alpha_{SA} = 0.1$, $\rho = 500$. in the optimization grid, PARADE reaches peak overall performance *and* fairness (measured by low subgroup gap and high performance for the worst-performing subgroup) simultaneously. Thus, we could conclude that this choice of $\alpha_{SA}$ and $\rho$ represents an optimal tradeoff for utility and fairness

| Color | overall | overall | black | black | blue | blue | brown | brown |
|---|---|---|---|---|---|---|---|---|
| **Method** | Parade | Margin (D) | Parade | Margin (D) | Parade | Margin (D) | Parade | Margin (D) |
| Recall@1 | $0.785 \pm 0.003$ | $0.786 \pm 0.003$ | $0.780 \pm 0.006$ | $0.777 \pm 0.005$ | $0.837 \pm 0.016$ | $0.841 \pm 0.010$ | $0.773 \pm 0.005$ | $0.779 \pm 0.008$ |
| NMI | $0.860 \pm 0.001$ | $0.861 \pm 0.003$ | $0.832 \pm 0.005$ | $0.837 \pm 0.004$ | $0.840 \pm 0.025$ | $0.863 \pm 0.026$ | $0.831 \pm 0.013$ | $0.838 \pm 0.007$ |
| $U_{KL}$ | $0.071 \pm 0.002$ | $0.076 \pm 0.001$ | $0.164 \pm 0.002$ | $0.153 \pm 0.003$ | $0.296 \pm 0.004$ | $0.275 \pm 0.007$ | $0.156 \pm 0.004$ | $0.148 \pm 0.003$ |
| Precision | $0.819 \pm 0.003$ | $0.822 \pm 0.001$ | $0.403 \pm 0.007$ | $0.412 \pm 0.016$ | $0.399 \pm 0.053$ | $0.401 \pm 0.022$ | $0.396 \pm 0.005$ | $0.390 \pm 0.016$ |
| Recall | $0.812 \pm 0.003$ | $0.816 \pm 0.002$ | $0.375 \pm 0.007$ | $0.384 \pm 0.014$ | $0.367 \pm 0.052$ | $0.366 \pm 0.022$ | $0.357 \pm 0.016$ | $0.351 \pm 0.016$ |
| Accuracy | $0.812 \pm 0.003$ | $0.815 \pm 0.002$ | $0.790 \pm 0.009$ | $0.798 \pm 0.004$ | $0.867 \pm 0.009$ | $0.877 \pm 0.011$ | $0.804 \pm 0.001$ | $0.812 \pm 0.006$ |

| Color | buff | buff | green | green | grey | grey | iridescent | iridescent | olive |
|---|---|---|---|---|---|---|---|---|---|
| **Method** | Parade | Margin (D) | Parade | Margin (D) | Parade | Margin (D) | Parade | Margin (D) | Parade |
| Recall@1 | $0.787 \pm 0.008$ | $0.792 \pm 0.006$ | $1.000 \pm 0.000$ | $1.000 \pm 0.000$ | $0.751 \pm 0.015$ | $0.754 \pm 0.010$ | $1.000 \pm 0.000$ | $1.000 \pm 0.000$ | $0.589 \pm 0.038$ |
| NMI | $0.789 \pm 0.005$ | $0.808 \pm 0.015$ | $-0.000 \pm 0.000$ | $-0.000 \pm 0.000$ | $0.853 \pm 0.009$ | $0.849 \pm 0.004$ | $1.000 \pm 0.000$ | $0.800 \pm 0.447$ | $0.164 \pm 0.008$ |
| $U_{KL}$ | $0.349 \pm 0.003$ | $0.343 \pm 0.004$ | $0.262 \pm 0.024$ | $0.252 \pm 0.021$ | $0.142 \pm 0.005$ | $0.137 \pm 0.002$ | $0.369 \pm 0.013$ | $0.318 \pm 0.017$ | $0.164 \pm 0.008$ |
| Precision | $0.250 \pm 0.006$ | $0.253 \pm 0.013$ | $1.000 \pm 0.000$ | $1.000 \pm 0.000$ | $0.337 \pm 0.014$ | $0.338 \pm 0.022$ | $1.000 \pm 0.000$ | $1.000 \pm 0.000$ | $0.289 \pm 0.077$ |
| Recall | $0.209 \pm 0.005$ | $0.212 \pm 0.012$ | $1.000 \pm 0.000$ | $1.000 \pm 0.000$ | $0.292 \pm 0.013$ | $0.297 \pm 0.021$ | $1.000 \pm 0.000$ | $1.000 \pm 0.000$ | $0.173 \pm 0.048$ |
| Accuracy | $0.824 \pm 0.003$ | $0.824 \pm 0.011$ | $1.000 \pm 0.000$ | $1.000 \pm 0.000$ | $0.790 \pm 0.007$ | $0.796 \pm 0.006$ | $1.000 \pm 0.000$ | $1.000 \pm 0.000$ | $0.600 \pm 0.033$ |

| Color | olive | orange | orange | red | red | white | white | yellow | yellow |
|---|---|---|---|---|---|---|---|---|---|
| **Method** | Margin (D) | Parade | Margin (D) | Parade | Margin (D) | Parade | Margin (D) | Parade | Margin (D) |
| Recall@1 | $0.620 \pm 0.087$ | $0.839 \pm 0.010$ | $0.863 \pm 0.040$ | $0.917 \pm 0.018$ | $0.919 \pm 0.017$ | $0.729 \pm 0.012$ | $0.706 \pm 0.011$ | $0.846 \pm 0.003$ | $0.862 \pm 0.012$ |
| NMI | $0.000 \pm 0.000$ | $0.701 \pm 0.060$ | $0.657 \pm 0.036$ | $0.839 \pm 0.032$ | $0.830 \pm 0.033$ | $0.792 \pm 0.008$ | $0.787 \pm 0.010$ | $0.842 \pm 0.011$ | $0.864 \pm 0.005$ |
| $U_{KL}$ | $0.151 \pm 0.005$ | $0.295 \pm 0.004$ | $0.277 \pm 0.009$ | $0.445 \pm 0.003$ | $0.411 \pm 0.011$ | $0.176 \pm 0.003$ | $0.166 \pm 0.003$ | $0.215 \pm 0.005$ | $0.202 \pm 0.004$ |
| Precision | $0.240 \pm 0.063$ | $0.302 \pm 0.027$ | $0.303 \pm 0.079$ | $0.555 \pm 0.052$ | $0.559 \pm 0.048$ | $0.387 \pm 0.018$ | $0.380 \pm 0.012$ | $0.503 \pm 0.012$ | $0.530 \pm 0.022$ |
| Recall | $0.160 \pm 0.049$ | $0.261 \pm 0.024$ | $0.263 \pm 0.076$ | $0.542 \pm 0.053$ | $0.544 \pm 0.049$ | $0.357 \pm 0.018$ | $0.342 \pm 0.011$ | $0.481 \pm 0.011$ | $0.509 \pm 0.022$ |
| Accuracy | $0.660 \pm 0.060$ | $0.867 \pm 0.017$ | $0.860 \pm 0.028$ | $0.923 \pm 0.017$ | $0.927 \pm 0.009$ | $0.752 \pm 0.004$ | $0.737 \pm 0.002$ | $0.879 \pm 0.001$ | $0.884 \pm 0.008$ |

Table 13: *Absolute performance for all CUB200 subgroups.* Metrics over each bird color subgroup in the CUB200 test dataset respectively, in representation space and downstream classification (logistic regressor) over 3 seeds for standard methods and PARADE in CUB200.

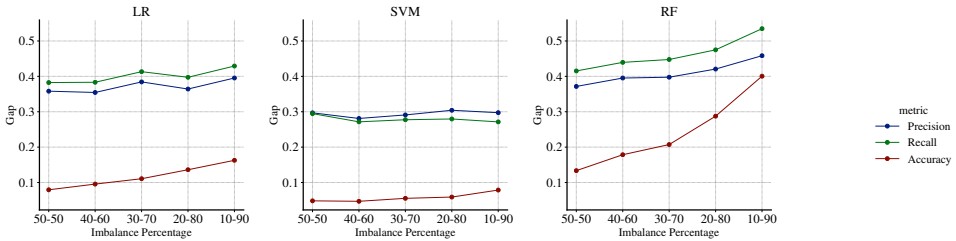

Figure 9: Impact of varying imbalance between the *minoritized* and *majoritized* classes on various downstream classifiers (RF, LR and SVM) in the manually class imbalanced CUB200 experiments. (Note: the imbalance percentage $50 - 50$ is equivalent to the balanced setting). Gaps increase for all classifiers downstream with more imbalance introduced to the upstream training data.

| | | | Overall | | | |
|---|---|---|---|---|---|---|
| | | Arcface | | Margin · Distance | | N-Pair · N-Pair |
| | | PARADE | Standard | PARADE | Standard | Standard |
| UPSTREAM EMBEDDING | Recall@1 | $0.897 \pm 0.002$ | $0.888 \pm 0.002$ | $0.885 \pm 0.002$ | $0.922 \pm 0.001$ | $0.11 \pm 0.002$ |
| | NMI | $0.91 \pm 0.0$ | $0.902 \pm 0.001$ | $0.901 \pm 0.003$ | $0.929 \pm 0.0$ | $0.61 \pm 0.0$ |
| | $U_{KL}$ | $0.019 \pm 0.001$ | $0.017 \pm 0.0$ | $0.336 \pm 0.013$ | $0.237 \pm 0.006$ | $2.595 \pm 0.055$ |
| DOWNSTREAM CLASSIFICATION LR | Accuracy | $0.891 \pm 0.0$ | $0.89 \pm 0.001$ | $0.692 \pm 0.004$ | $0.831 \pm 0.002$ | $0.017 \pm 0.0$ |
| | Precision | $0.721 \pm 0.0$ | $0.721 \pm 0.001$ | $0.55 \pm 0.003$ | $0.652 \pm 0.003$ | $0.004 \pm 0.0$ |
| | Recall | $0.74 \pm 0.0$ | $0.741 \pm 0.001$ | $0.546 \pm 0.003$ | $0.674 \pm 0.003$ | $0.011 \pm 0.0$ |

Table 14: *Overall results on CelebA.* Metrics over entire test dataset in representation space and downstream classification (logistic regressor) over 3 seeds for standard methods and PARADE in CelebA.

in PARADE as measured by Recall@1. By the other displayed metrics, we see that $\alpha_{SA} = 0.1$, $\rho = 500$. demonstrates a reasonable utility-fairness tradeoff. Therefore, the choice of $\alpha_{SA} = 0.1$, $\rho = 500$. would be optimal for PARADE in CUB200 bird color setting. Note that the choice of where to operate within this trade-off should depend on the application that is being targeted. For

|  |  | Arcface | | Subgroup Gap
Margin · Distance | | N-Pair · N-Pair |
|  |  | PARADE | Standard | PARADE | Standard | Standard |
|---|---|---|---|---|---|---|
| UPSTREAM EMBEDDING | Recall@1 | $0.135 \pm 0.008$ | $0.128 \pm 0.003$ | $0.085 \pm 0.009$ | $0.122 \pm 0.005$ | $-0.023 \pm 0.013$ |
|  | NMI | $-0.003 \pm 0.004$ | $-0.01 \pm 0.002$ | $-0.012 \pm 0.003$ | $-0.002 \pm 0.003$ | $-0.102 \pm 0.002$ |
|  | $U_{KL}$ | $-0.054 \pm 0.003$ | $-0.052 \pm 0.003$ | $-0.04 \pm 0.011$ | $-0.03 \pm 0.007$ | $-0.015 \pm 0.038$ |
| DOWNSTREAM CLASSIFICATION | LR Accuracy | $0.068 \pm 0.002$ | $0.069 \pm 0.002$ | $0.131 \pm 0.006$ | $0.082 \pm 0.005$ | $0.006 \pm 0.002$ |
|  | Precision | $0.087 \pm 0.003$ | $0.087 \pm 0.004$ | $0.146 \pm 0.006$ | $0.1 \pm 0.007$ | $0.001 \pm 0.001$ |
|  | Recall | $0.084 \pm 0.002$ | $0.083 \pm 0.003$ | $0.141 \pm 0.007$ | $0.098 \pm 0.007$ | $0.002 \pm 0.001$ |

Table 15: *Gap study on CelebA.* Average gaps in representation space and downstream classification (logistic regressor) over 3 seeds between minoritized and majoritized classes (Fitzpatrick Skintone) for standard methods and PARADE in CelebA.

|  |  | Margin · Distance | |
|  |  | PARADE | Standard |
|---|---|---|---|
| UPSTREAM EMBEDDING | Recall@1 | $-0.035 \pm 0.006$ | $0.005 \pm 0.004$ |
|  | NMI | $-0.004 \pm 0.003$ | $0.004 \pm 0.002$ |
|  | $U_{KL}$ | $0.04 \pm 0.011$ | $0.084 \pm 0.006$ |
| DOWNSTREAM CLASSIFICATION (LR) | Precision | $0.021 \pm 0.006$ | $0.039 \pm 0.002$ |
|  | Recall | $0.018 \pm 0.006$ | $0.029 \pm 0.002$ |
|  | Accuracy | $0.011 \pm 0.005$ | $0.018 \pm 0.002$ |

Table 16: *Gap study on CelebA excluding Fitzpatrick Skintone VI.* Average gaps in representation space and downstream classification (logistic regressor) over 3 seeds between minoritized and majoritized classes (Fitzpatrick Skintone) where the darkest skintone (VI) is excluded for standard methods and PARADE in CelebA.

| **Skintones** | Overall | Overall | Skintone 1 | Skintone 1 | Skintone 2 | Skintone 2 | Skintone 3 |
| **Method** | Parade | Margin (D) | Parade | Margin (D) | Parade | Margin (D) | Parade |
|---|---|---|---|---|---|---|---|
| Recall@1 | $0.885 \pm 0.002$ | $0.922 \pm 0.001$ | $0.738 \pm 0.005$ | $0.858 \pm 0.011$ | $0.887 \pm 0.004$ | $0.930 \pm 0.001$ | $0.907 \pm 0.001$ |
| NMI@1 | $0.901 \pm 0.003$ | $0.929 \pm 0.000$ | $0.933 \pm 0.007$ | $0.961 \pm 0.000$ | $0.923 \pm 0.004$ | $0.947 \pm 0.001$ | $0.927 \pm 0.002$ |
| $U_{KL}$ | $0.336 \pm 0.013$ | $0.237 \pm 0.006$ | $0.436 \pm 0.020$ | $0.419 \pm 0.002$ | $0.350 \pm 0.014$ | $0.260 \pm 0.004$ | $0.350 \pm 0.012$ |
| Precision | $0.550 \pm 0.003$ | $0.652 \pm 0.003$ | $0.398 \pm 0.011$ | $0.639 \pm 0.005$ | $0.566 \pm 0.003$ | $0.696 \pm 0.002$ | $0.558 \pm 0.003$ |
| Recall | $0.546 \pm 0.003$ | $0.674 \pm 0.003$ | $0.421 \pm 0.014$ | $0.656 \pm 0.005$ | $0.604 \pm 0.003$ | $0.739 \pm 0.003$ | $0.578 \pm 0.003$ |
| Accuracy | $0.692 \pm 0.004$ | $0.831 \pm 0.002$ | $0.578 \pm 0.009$ | $0.783 \pm 0.004$ | $0.707 \pm 0.004$ | $0.843 \pm 0.002$ | $0.716 \pm 0.004$ |

| **Skintones** | Skintone 3 | Skintone 4 | Skintone 4 | Skintone 5 | Skintone 5 | Skintone 6 | Skintone 6 |
| **Method** | Margin (D) | Parade | Margin (D) | Parade | Margin (D) | Parade | Margin (D) |
|---|---|---|---|---|---|---|---|
| Recall@1 | $0.937 \pm 0.003$ | $0.850 \pm 0.002$ | $0.893 \pm 0.003$ | $0.785 \pm 0.017$ | $0.838 \pm 0.004$ | $0.642 \pm 0.018$ | $0.628 \pm 0.006$ |
| NMI@1 | $0.947 \pm 0.001$ | $0.927 \pm 0.002$ | $0.946 \pm 0.000$ | $0.943 \pm 0.002$ | $0.957 \pm 0.004$ | $0.948 \pm 0.004$ | $0.960 \pm 0.009$ |
| $U_{KL}$ | $0.241 \pm 0.006$ | $0.339 \pm 0.012$ | $0.240 \pm 0.008$ | $0.371 \pm 0.012$ | $0.288 \pm 0.013$ | $0.547 \pm 0.010$ | $0.483 \pm 0.014$ |
| Precision | $0.672 \pm 0.003$ | $0.471 \pm 0.005$ | $0.644 \pm 0.003$ | $0.355 \pm 0.011$ | $0.570 \pm 0.001$ | $0.258 \pm 0.005$ | $0.492 \pm 0.021$ |
| Recall | $0.708 \pm 0.003$ | $0.518 \pm 0.005$ | $0.695 \pm 0.002$ | $0.386 \pm 0.011$ | $0.602 \pm 0.001$ | $0.275 \pm 0.005$ | $0.511 \pm 0.019$ |
| Accuracy | $0.842 \pm 0.003$ | $0.632 \pm 0.005$ | $0.798 \pm 0.002$ | $0.545 \pm 0.009$ | $0.747 \pm 0.002$ | $0.430 \pm 0.010$ | $0.678 \pm 0.015$ |

Table 17: *Absolute performance for all CelebA subgroups.* Metrics over each Fitzpatrick Skintone subgroup in the CelebA test dataset respectively, in representation space and downstream classification (logistic regressor) over 3 seeds for standard methods and PARADE in CelebA.

example, here we use Recall@1 to determine the optimal choice of hyperparameters and validate with the other two considered metrics. However, for LFW, which has a high population of singleton classes (see Figure 7), NMI would be a better metric to use for selecting optimal point.

# D    IMPLEMENTATION DETAILS

## D.1    DATASET ATTRIBUTE INFORMATION

Dataset manipulation for the CARS196 and CUB200 manually class imbalanced experimentsis explained in Section 3.3.

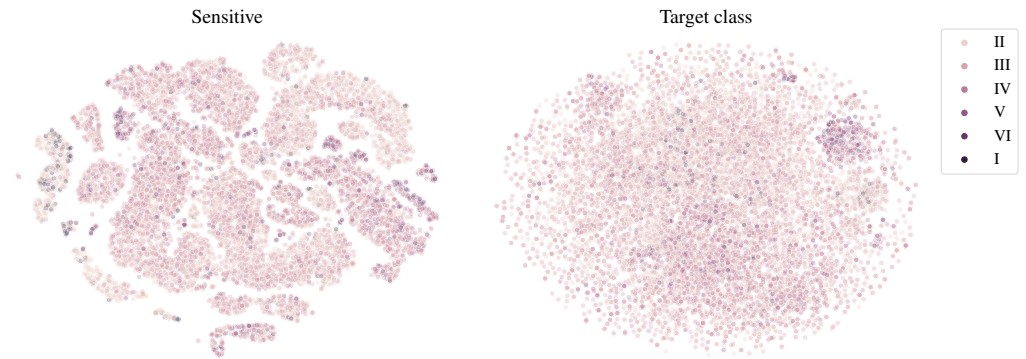

Figure 10: A t-SNE (Maaten & Hinton, 2008) visualization of the two distinct PARADE embeddings for Fitzpatrick Skintone CelebA experiments: the sensitive attribute embedding (**left**) and the class label embedding (**right**).

| | | Arcface | | Overall
Margin · Distance | | N-Pair · N-Pair |
|---|---|---|---|---|---|---|
| | | PARADE | Standard | PARADE | Standard | Standard |
| UPSTREAM
EMBEDDING | Recall@1 | $0.268 \pm 0.01$ | $0.306 \pm 0.008$ | $0.329 \pm 0.002$ | $0.381 \pm 0.004$ | $0.187 \pm 0.006$ |
| | NMI | $0.849 \pm 0.005$ | $0.859 \pm 0.002$ | $0.865 \pm 0.001$ | $0.869 \pm 0.001$ | $0.854 \pm 0.001$ |
| | $U_{KL}$ | $0.118 \pm 0.021$ | $0.089 \pm 0.014$ | $0.129 \pm 0.001$ | $0.103 \pm 0.001$ | $1.815 \pm 0.011$ |
| DOWNSTREAM
CLASSIFICATION RF | Accuracy | $0.8 \pm 0.0$ | $0.804 \pm 0.017$ | $0.762 \pm 0.002$ | $0.8 \pm 0.003$ | $0.887 \pm 0.002$ |
| | Precision | $0.789 \pm 0.003$ | $0.793 \pm 0.016$ | $0.767 \pm 0.001$ | $0.788 \pm 0.004$ | $0.878 \pm 0.003$ |
| | Recall | $0.827 \pm 0.0$ | $0.831 \pm 0.015$ | $0.801 \pm 0.003$ | $0.823 \pm 0.003$ | $0.913 \pm 0.002$ |

Table 18: *Overall results on LFW.* Metrics over entire test dataset in representation space and downstream classification (random forest) over 3 seeds for standard methods and PARADE in LFW. **Note**: Due to the number of singleton classes in LFW, Recall@1 is not considered a good metric of performance for this dataset.

| | | Arcface | | Subgroup Gap
Margin · Distance | | N-Pair · N-Pair |
|---|---|---|---|---|---|---|
| | | PARADE | Standard | PARADE | Standard | Standard |
| UPSTREAM
EMBEDDING | Recall@1 | $0.039 \pm 0.017$ | $0.061 \pm 0.017$ | $0.075 \pm 0.014$ | $0.068 \pm 0.013$ | $0.054 \pm 0.01$ |
| | NMI | $0.048 \pm 0.011$ | $0.057 \pm 0.003$ | $0.041 \pm 0.003$ | $0.048 \pm 0.003$ | $0.048 \pm 0.003$ |
| | $U_{KL}$ | $0.176 \pm 0.019$ | $0.157 \pm 0.011$ | $0.163 \pm 0.003$ | $0.165 \pm 0.005$ | $0.357 \pm 0.012$ |
| DOWNSTREAM
CLASSIFICATION RF | Accuracy | $0.04 \pm 0.01$ | $0.038 \pm 0.012$ | $0.049 \pm 0.005$ | $0.038 \pm 0.005$ | $0.025 \pm 0.004$ |
| | Precision | $0.036 \pm 0.018$ | $0.041 \pm 0.014$ | $0.04 \pm 0.005$ | $0.037 \pm 0.007$ | $0.025 \pm 0.007$ |
| | Recall | $0.066 \pm 0.017$ | $0.076 \pm 0.015$ | $0.066 \pm 0.006$ | $0.071 \pm 0.007$ | $0.041 \pm 0.006$ |

Table 19: *Gap study on LFW.* Average gaps in representation space and downstream classification (random forest) over 3 seeds between minoritized and majoritized classes (Race) for standard methods and PARADE in LFW.

| | | Asian | | Black | | Indian
Margin · Distance | | White | |
|---|---|---|---|---|---|---|---|---|---|
| | | PARADE | Standard | PARADE | Standard | PARADE | Standard | PARADE | Standard |
| UPSTREAM
EMBEDDING | Recall@1 | $0.262 \pm 0.028$ | $0.31 \pm 0.014$ | $0.289 \pm 0.011$ | $0.331 \pm 0.016$ | $0.238 \pm 0.027$ | $0.325 \pm 0.032$ | $0.338 \pm 0.004$ | $0.39 \pm 0.004$ |
| | NMI | $0.894 \pm 0.003$ | $0.914 \pm 0.005$ | $0.858 \pm 0.001$ | $0.882 \pm 0.005$ | $0.948 \pm 0.007$ | $0.951 \pm 0.007$ | $0.862 \pm 0.001$ | $0.868 \pm 0.0$ |
| | $U_{KL}$ | $0.304 \pm 0.003$ | $0.265 \pm 0.003$ | $0.456 \pm 0.009$ | $0.417 \pm 0.013$ | $0.141 \pm 0.002$ | $0.133 \pm 0.005$ | $0.137 \pm 0.002$ | $0.107 \pm 0.001$ |
| DOWNSTREAM
CLASSIFICATION RF | Accuracy | $0.81 \pm 0.007$ | $0.828 \pm 0.008$ | $0.827 \pm 0.005$ | $0.853 \pm 0.007$ | $0.772 \pm 0.011$ | $0.814 \pm 0.012$ | $0.754 \pm 0.002$ | $0.794 \pm 0.003$ |
| | Precision | $0.715 \pm 0.012$ | $0.726 \pm 0.013$ | $0.743 \pm 0.008$ | $0.759 \pm 0.014$ | $0.664 \pm 0.006$ | $0.711 \pm 0.01$ | $0.747 \pm 0.002$ | $0.769 \pm 0.004$ |
| | Recall | $0.713 \pm 0.013$ | $0.72 \pm 0.014$ | $0.751 \pm 0.008$ | $0.758 \pm 0.01$ | $0.657 \pm 0.008$ | $0.702 \pm 0.011$ | $0.773 \pm 0.002$ | $0.798 \pm 0.003$ |

Table 20: *Absolute performance for all LFW subgroups.* Metrics over each Race subgroup in the LFW test dataset respectively, in representation space and downstream classification (random forest) over 3 seeds for standard methods and PARADE in LFW.

For CUB200 bird color experiments, we utilized the labeled bird color attributes from Wah et al. (2011). Each image can have multiple "primary color" labels. Therefore, we take the mode over

Table 21: *Benchmarking additional fairness improvement methods in downstream classification on LFW.* Overall performance and subgroup gaps for Domain-Independent Training and Oversampling (Wang et al., 2020b) on LFW with Race attribute.

(a) Domain-Independent Training

| METRIC ↓ | ArcFace | Margin (D) | N-Pair |
|---|---|---|---|
| **Overall** | | | |
| ACCURACY | $0.759 \pm 0.017$ | $0.753 \pm 0.002$ | $0.861 \pm 0.005$ |
| PRECISION | $0.793 \pm 0.010$ | $0.786 \pm 0.003$ | $0.872 \pm 0.003$ |
| RECALL | $0.799 \pm 0.012$ | $0.792 \pm 0.003$ | $0.889 \pm 0.003$ |
| **Gap** | | | |
| ACCURACY | $0.093 \pm 0.011$ | $0.095 \pm 0.006$ | $0.029 \pm 0.009$ |
| PRECISION | $0.030 \pm 0.011$ | $0.020 \pm 0.009$ | $0.029 \pm 0.010$ |
| RECALL | $0.040 \pm 0.012$ | $0.044 \pm 0.008$ | $0.026 \pm 0.010$ |

(b) Oversampling

| METRIC ↓ | ArcFace | Margin (D) | N-Pair |
|---|---|---|---|
| **Overall** | | | |
| ACCURACY | $0.775 \pm 0.017$ | $0.767 \pm 0.004$ | $0.881 \pm 0.004$ |
| PRECISION | $0.771 \pm 0.014$ | $0.762 \pm 0.002$ | $0.873 \pm 0.005$ |
| RECALL | $0.815 \pm 0.014$ | $0.807 \pm 0.002$ | $0.909 \pm 0.004$ |
| **Gap** | | | |
| ACCURACY | $0.070 \pm 0.012$ | $0.072 \pm 0.006$ | $0.032 \pm 0.006$ |
| PRECISION | $0.020 \pm 0.013$ | $0.024 \pm 0.009$ | $0.027 \pm 0.009$ |
| RECALL | $0.023 \pm 0.013$ | $0.031 \pm 0.011$ | $0.036 \pm 0.008$ |

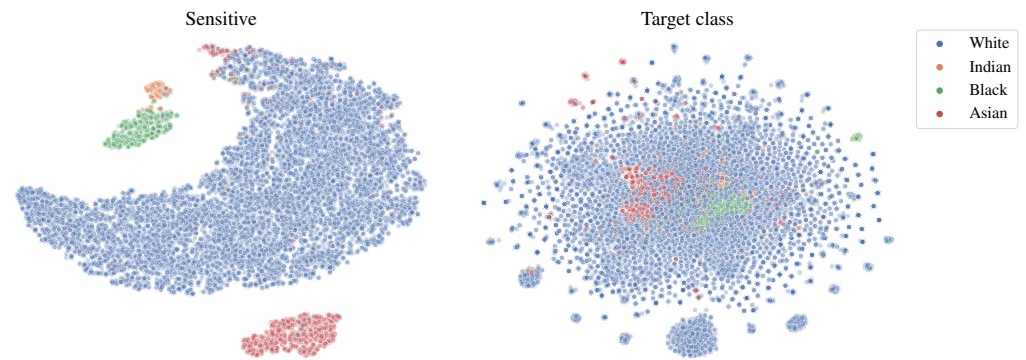

Figure 11: A t-SNE (Maaten & Hinton, 2008) visualization of the two distinct PARADE embeddings for Race LFW experiments: the sensitive attribute embedding (**left**) and the class label embedding (**right**).

| Dataset | Protected Attribute | Protected Attribute Values |
|---|---|---|
| CUB200-2011 | Color | Black, Blue, Brown, Buff, Green, Grey, Iridescent Olive, Orange, Red, White, Yellow |
| CelebA | Fitzpatrick Skintone Category | I, II, III, IV, V, VI |
| LFW | Race | Asian, Black, Indian, White |

Table 22: *Summarizing attribute information.* Protected attribute examined and associated values taken by the protected attribute in each dataset analyzed w.r.t. a sensitive attribute in the main paper (CUB200, CelebA, LFW).

all bird colors labeled for each image in order to determine a single bird color associated with the image. For CelebA, we calculate the Fitzpatrick skintone based on the image pixel information for each image. The calculation is described in Section D.2. For LFW, we construct the "Race" attribute from labels of "White", "Black", "Asian," and "Indian" as labelled by Kumar et al. (2009). For each of these attributes, the labelling provided by Kumar et al. (2009) has a float value, which we map to binary values: the image is considered to have the attribute if the value is greater than 0, and the image is considered to not have the attribute if the value is less than 0. Naturally, the labelling is not necessarily correct for each image, as the confidence about the "Race" labelling can be quite low for some images. We remove all images without at least one of these attributes, though we note that these attributes do not encompass all races. Therefore, our analysis may not be relevant for other races not labelled by Kumar et al. (2009).

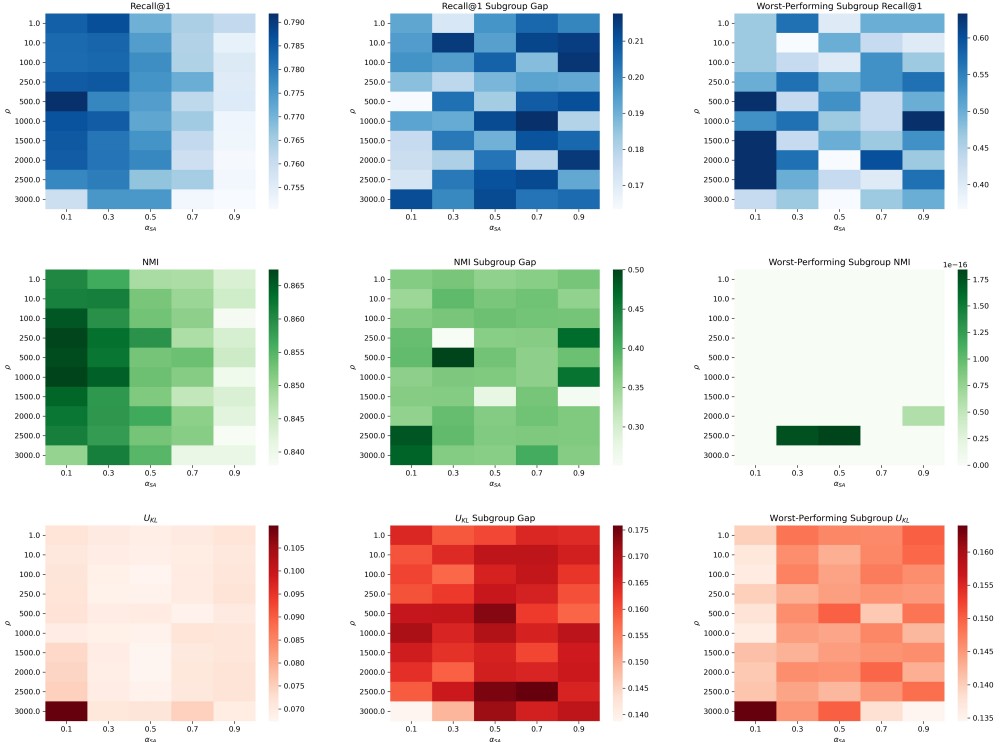

Figure 12: *Exploring fairness-utility tradeoffs in PARADE on CUB200 over grid of $\alpha_{SA}$ and $\rho$ values.* Overall performance (left column), subgroup gap (middle column) and worst-group performance (right column) over metrics Recall@1 (top row), NMI (middle row), and $U_{KL}$ (bottom row) in PARADE on CUB200. $\alpha_{SA}$ and $\rho$ in PARADE objective (Section 4) varied from 0.1 to 0.9, and 1 to 3000, respectively.

## D.2 FITZPATRICK SKINTONE CALCULATION

We follow the methods from Cheng et al. (2021) for calculation of Fitzpatrick Skintone based on image pixel information. However, we calculate these values for CelebA, as opposed to CelebA-HQ. As CelebA-HQ incorporates higher resolution images, but has fewer images, our process of Fitzpatrick Skintone calculation on CelebA is slightly modified to account for lower resolution, and differing image size.

In Cheng et al. (2021), two sample skin patches are selected from each image of CelebA-HQ to determine the skintone. We select three sample skin patches, as we are forced to reduce the dimensions of the patches to account for the smaller image size of CelebA. Additionally, we leverage facial landmark attributes provided by CelebA Liu et al. (2015) in order to choose our sample patches. Specifically, given the $(x, y)$ landmarks for the left eye, right eye, and nose for each image, we choose to sample square patches of size $20 \times 20$ (all 3 color channels are selected) with the following center points:

$$(x_{\text{left eye}}, y_{\text{nose}})$$
$$(x_{\text{right eye}}, y_{\text{nose}})$$
$$(x_{\text{nose}}, y_{\text{nose}})$$

The first two center points are intended to capture the likely location of the *left* and *right* cheeks, respectively, as these are likely located below each eye and adjacent to the nose. The last center point is the nose. We note that this protected attribute generation is not perfect. In some cases, such label generation can accidentally use aspects of the background, if, the individual's face position in the image is not facing forward. Also, extreme lighting can lead to misclassification of skintone. Nonetheless, we believe the procedure provides a good approximation of Fitzpatrick skintone category, but do not recommend these attribute labels for use outside of fairness analysis.

The selected sample patches are converted to CIELab-space to retrieve the $L$ (luminance) and $b$ (yellow) values. We then calculate the Mean Individual Typology Angle (ITA) value:

$$ITA = \arctan\left(\frac{L - 50}{b}\right) \times \frac{180°}{\pi}$$

Table 23: Fitzpatrick Skin Tone Categories corresponding to Mean ITA values, information taken from Cheng et al. (2021)

| ITA Range | Fitzpatrick Category | Description |
|---|---|---|
| $50 \leq ITA$ | I | Extremely Light |
| $40 \leq ITA < 50$ | II | Very Light |
| $30 \leq ITA < 40$ | III | Light / Somewhat Light |
| $20 \leq ITA < 30$ | IV | Dark / Somewhat Dark |
| $10 \leq ITA < 20$ | V | Very Dark |
| $ITA < 10$ | VI | Extremely Dark |

Based on the Mean ITA calculation, we classify each image into one of the 6 Fitzpatrick skintone categories, as listed in Table 23. To calculate subgroup gaps, we calculate gaps between the mean value over the 3 lightest Fitzpatrick skintones and the mean value over the 3 darkest Fitzpatrick skintones.

### D.3 TRAINING PARAMETERS

For CUB200 and CARS196, we did not perform hyperparameter search but followed reported hyperparameters from Roth et al. (2020c) for best performance with an ImageNet Deng et al. (2009) pretrained ResNet50 He et al. (2016) and frozen batch normalization layers. As detailed in Roth et al. (2020c), we train for 150 epochs with embedding dimension 128, learning rate 0.00001 with no scheduler, and weight decay 0.0004. We train with a batch size of 128, with the Adam optimizer Kingma & Ba (2015) over five seeds inclusive for the balance control datasets, and for CUB200 color experiments; and seeds $0 - 9$ for the manually class imbalanced experiments. For training transforms, we normalize each image using color channel means $(0.485, 0.456, 0.406)$ and standard deviations $(0.229, 0.224, 0.225)$, randomly crop the image and re-size to $224 \times 224$ and horizontally flip with probability 0.5. For testing transforms, we normalize each image with the aforementioned color channel means and standard deviations, resize to $256 \times 256$, and center crop to $224 \times 224$.

For CelebA and LFW, we performed hyperparameter search over the following hyperparameters: **architectures**: ResNet50 He et al. (2016), and SE-Net50 (both with and without frozen batch normalization layers); **number of training epochs**; **learning rates**; **last linear layer learning rate** (differ from other layer learning rates); **learning rate schedulers**; **embedding dimensions**: 64, 128, 256; **pre-training**; **image augmentations**. We evaluated hyperparameter sets on a validation set we cut from the typical training set (20% of training set), and chose the set of hyperparameters with best recall@k score for CelebA and best NMI score for LFW. NMI is used for LFW due to the high number of singleton classes present in the dataset (recall@1 is meaningless for singleton classes).

For CelebA, we train on the ResNet50 He et al. (2016) architecture with frozen batch normalization layers, for 125 epochs with learning rate 0.00001, and no scheduler, weight decay 0.0004, Adam Kingma & Ba (2015) optimizer, and batch size of 128. For training transforms, we normalize each image using color channel means $(0.5, 0.5, 0.5)$ and standard deviations $(0.5, 0.5, 0.5)$, resize to $256 \times 256$, center crop to $224 \times 224$ and horizontally flip with probability 0.5. For testing transforms, we normalize each image with the aforementioned color channel means and standard deviations, resize to $256 \times 256$, and center crop to $224 \times 224$. We average over runs with seeds $0 - 2$, inclusive.

For LFW, we train on the ResNet50 He et al. (2016) architecture with frozen batch normalization layers, for 125 epochs with initial learning rate 0.00001 for all model parameters except the last linear layer, which has initial learning rate 0.0001, and a multi-step learning rate scheduler which reduces the learning rate by a factor of 0.3 at epochs 50 and 100, weight decay 0.0004, Adam Kingma & Ba

(2015) optimizer, and batch size of $64$. For training transforms, we normalize each image using color channel means $(0.5, 0.5, 0.5)$ and standard deviations $(0.5, 0.5, 0.5)$, resize to $256 \times 256$, center crop to $224 \times 224$ and horizontally flip with probability $0.5$. For testing transforms, we normalize each image with the aforementioned color channel means and standard deviations, resize to $256 \times 256$, and center crop to $224 \times 224$. We average over runs with seeds $0 - 2$, inclusive.

For each dataset we chose a set of loss and batch mining strategies that have historically been used for the relevant task, encompassing a broad range of methods, and / or achieved good performance. However, for n-pair loss and sampling, good performance was not achieved for the facial datasets despite use in the past for facial recognition Sohn (2016). For manually class imbalanced experiments with CARS196 and CUB200 and the associated balanced controls, we used: margin loss / distance-weighted sampling, margin loss / semi-hard sampling, triplet loss / distance-weighted sampling, triplet loss / semi-hard sampling, contrastive loss / distance-weighted sampling, multisimilarity loss, and proxy-NCA loss. For the color experiments with CUB200, we used: margin loss / distance-weighted sampling. For CelebA and LFW, we used: margin loss / distance-weighted sampling, arcface loss, and n-pair loss and sampling. For all testing and evaluation experiments with PARADE, we used margin loss and distance-weighted sampling, but PARADE can be used with any loss and mining strategy.

**DML-specific parameters**   Here we list the hyperparameters that we use for each evaluated loss function and batch mining strategy, if applicable. Refer to A for explicit formulas associated with the parameters here. We set $\gamma = 0.2$ in semi-hard mining. For distance-weighted mining, we set $\lambda = 0.5$ and clip the maximum distance to $1.4$. In the triplet objective, we use $\gamma = 0.2$ for triplet loss. For margin loss, the learning rate of the boundary $\beta$ is set to $0.0005$, with initial value $1.2$ and triplet margin $\gamma = 0.2$. For N-Pair uses embedding regularization parameter $\nu = 0.005$. In Multisimilarity loss, we use $\alpha = 2$, $\beta = 40$, $\lambda = 0.5$ and $\epsilon = 0.1$. Finally, for ArcFace, additive angular margin penalty is set to $\gamma = 0.5$, while scaling parameter $s = 16$ and class centers are optimized with learning rate $0.0005$.

The two PARADE parameters, $\alpha_{SA}$ and $\rho$, as described in Section 4, were optimized via worst-group performance over a grid search. For CUB200, we set $\alpha_{SA} = 0.3$, $\rho = 1500$. For CelebA, we set $\alpha_{SA} = 0.1$, $\rho = 1000$. For LFW, we set $\alpha_{SA} = 0.3$, $\rho = 100$.

### D.4   FAIRNESS EVALUATION

For each dataset, we calculate subgroup gaps between the *majoritized* and *minoritized* subgroup (CARS196, CUB200 *class*, CelebA) or between the worst-performing subgroup and other subgroups (LFW). In CUB200 *color* experiments, due to the large number of subgroups, we calculate the gap between the top 6 performing subgroups and the bottom 6 performing subgroups (there are 12 total subgroups).

**Upstream**   In the upstream embedding tasks, in which we denote $\phi$ as the embedding function for the learned model, and use $A(x)$ to denote the value of the attribute $A$ for data point $x$, we calculate recall@1 for data samples in $X$ with associated class label $Y$ and attribute $a \in A$ as:

$$\text{Recall@k} = \frac{1}{|\{x \in X : A(x) = a\}|} \sum_{\{x \in X : A(x) = a\}} \begin{cases} 1 & \exists \tilde{x} \in NN_k(x) : Y(\tilde{x}) = Y(x) \\ 0 & \text{else} \end{cases}$$

Note here that the nearest neighbors function is computed with respect to *all* $x \in X$, not exclusively $x \in X$ with attribute $a \in A$, but the input to the nearest neighbors function is exclusively $\{x \in X : A(x) = a\}$. To calculate NMI, let $\mathcal{C}$ be the output of a clustering algorithm $C$ on the entire dataset $X$, i.e. $\mathcal{C} = C(X)$ and let $\mathcal{C}|_{\mathcal{S}}$ denote the output of clustering algorithm $C$ restricted to some subset $\mathcal{S} \subset X$. The important note here is that the clustering algorithm is run over the entire dataset, but $\mathcal{C}|_{\mathcal{S}}$ expresses the cluster labels only for $\mathcal{S} \subset X$. Then, we measure NMI for data samples in $X$ with associated class label $Y$ and attribute $a \in A$ as:

$$NMI = \frac{I(Y(\{x \in X : A(x) = a\}); \mathcal{C}|_{\{x \in X : A(x) = a\}})}{H(Y(\{x \in X : A(x) = a\})) + H(\mathcal{C}|_{\{x \in X : A(x) = a\}})}$$

We calculate $U_{\text{KL}}$ for data samples in $X$ with attribute $a \in A$ as:

$$U_{\text{KL}}(X) = \mathcal{D}_{\text{KL}}\left(\mathcal{U}_{\text{D}}, \mathcal{S}_{\phi(\{x \in X : A(x) = a\})}\right)$$

where $\mathcal{S}_{\phi(\{x \in X : A(x) = a\})}$ denotes the singular values over $\phi(\{x \in X : A(x) = a\})$.

**Downstream** In the downstream tasks, for data samples in $X$ with class label $Y$, and predictor $\hat{Y}$, let $Y(x)$ express the value of the ground-truth label for data sample $x$ and let $\hat{Y}(x)$ express the value of the predicted label. Then, we denote $TP_a^{(y)}$ the number of true positives with attribute $a \in A$:

$$TP_a^{(y)} = \{x \in X : A(x) = a, Y(x) = y, \hat{Y}(x) = y\}$$

$FP_a^{(y)}$ the number of false positives with attribute $a \in A$

$$FP_a^{(y)} = \{x \in X : A(x) = a, Y(x) = y, \hat{Y}(x) \neq y\}$$

and $FN_a^{(y)}$ the number of false negatives with attribute $a \in A$:

$$FN_a^{(y)} = \{x \in X : A(x) = a, Y(x) \neq y, \hat{Y}(x) = y\}$$

for $y \in Y$.

We calculate macro-averaged recall for data samples in $X$ with associated class label $Y$ and attribute $a \in A$ as:

$$\text{Recall} = \frac{1}{|Y|} \sum_{y \in Y} \frac{TP_a^{(y)}}{TP_a^{(y)} + FN_a^{(y)}}$$

where $|Y|$ is the number of possible class labels, i.e. the size of the set of all possible values of $Y$. We calculate macro-averaged precision for data samples in $X$ with associated class label $Y$ and attribute $a \in A$ as:

$$\text{Precision} = \frac{1}{|Y|} \sum_{y \in Y} \frac{TP_a^{(y)}}{TP_a^{(y)} + FP_a^{(y)}}$$

We calculate accuracy for data samples in $X$ with associated class label $Y$ and attribute $a \in A$ as:

$$\text{Accuracy} = \frac{|\{x \in X : A(x) = a, Y(x) = \hat{Y}(x)\}|}{|\{x \in X : A(x) = a\}|}$$

The subgroup gaps are then considered to be the difference between the metric value for the *majoritized* subgroup and the metric value for the *minoritized* subgroup (CARS196, CUB200, CelebA); or between the metric value for *each subgroup with better performance than the worst-performing subgroup* and the metric value for the *worst-performing subgroup* (LFW). As stated in Section D.4, for CUB200 bird color experiments, the subgroup gaps were calculated between the top performing 50% of subgroups and bottom performing 50 of subgroups.

