# OpenReview forum: "Is Fairness Only Metric Deep? Evaluating and Addressing Subgroup Gaps in Deep Metric Learning"
_ICLR.cc/2022/Conference — ICLR 2022 Poster_

### Official Review · Reviewer_vBmc · 2021-10-31

**Correctness:** 3
**Technical Novelty And Significance:** 3
**Empirical Novelty And Significance:** 3
**Recommendation:** 6
**Confidence:** 4

**Main Review:**

Strengths:
1. The paper is well-written and easy to follow.
2. The paper studies an underexplored problem and proposes evaluation metrics to benchmark existing methods, which can inspire other researchers for future works.

Weaknesses:
1. The novelty of PARADE seems to be limited since the authors directly “use the adversarial separation (de-correlation) method from (Milbich et al., 2020)” (Subsec. “Partial De-correlation” in Sec. 4). Could the authors further demonstrate the novelty compared with (Milbich et al., 2020)? Otherwise, I do not think the authors can claim PARADE as “a novel objective” as one of the contributions (last paragraph of Sec. 1).
2. The authors show that the biases can be propagated to the downstream classification task and a naive method, training classifiers on the balanced dataset, cannot simply address the fairness issue. Since there are many bias mitigation methods ([1]) proposed for image classification, can authors also benchmark those bias mitigation methods for downstream classification? I understand that the authors want to pursue “fairness in representations” by proposing PARADE to address the fairness problem in the upstream embedding. However, I believe that the authors need to add some non-trivial bias mitigation methods to show that the bias cannot be fully addressed solely in the downstream stage, which can further demonstrate the necessity of achieving fairness in the upstream task, i.e., learning fairer representation in deep metric learning.

[1] Z. Wang et al., “Towards Fairness in Visual Recognition: Effective Strategies for Bias Mitigation,” in CVPR, 2020


**Summary Of The Paper:**

The paper investigates the fairness problem in the deep metric learning task, which is underexplored by the research community. The authors propose finDML to benchmark previous methods on multiple imbalanced datasets with three newly proposed metrics. The experimental results show that all previous deep metric learning methods have the fairness issue, i.e., larger performance gaps across different subgroups. The authors further propose PARADE to mitigate the biases, which has been shown to be effective by the experiments.

**Summary Of The Review:**

Since the paper has weaknesses in terms of the novelty of PARADE and missing experiments of bias mitigation in the downstream classification task, I recommend “marginally below the acceptance threshold” to this paper. I would like to increase my rating if the authors address my concerns.

============

After Authors' Response:

I appreciate the authors’ feedback, which addressed my concerns. I will increase my rating to “6: marginally above the acceptance threshold.”

---

> ### Author Response · Authors · 2021-11-19
> **PARADE explanation and additional results added for fairness improvement benchmarks downstream**
>
> Thank you for your thorough and constructive review of our work. We address the concerns highlighted in the review as follows and add additional experiments based on the review to the new upload of the paper:
>
> 1. > The novelty of PARADE seems to be limited since the authors directly “use the adversarial separation (de-correlation) method from (Milbich et al., 2020)” (Subsec. “Partial De-correlation” in Sec. 4). Could the authors further demonstrate the novelty compared with (Milbich et al., 2020)? Otherwise, I do not think the authors can claim PARADE as “a novel objective” as one of the contributions (last paragraph of Sec. 1).
>
> We have added additional differentiation between PARADE and existing adversarial separation from Milbich et al. in the first paragraph of Section 4 in our revision. We include the added text here for reference:
>
> ***​​We enumerate several significant changes: 1) only target embedding released at test-time; 2) triplet formation and loss term w.r.t. sensitive attribute; 3) de-correlation with sensitive attribute as opposed to de-correlation to reduce redundancy in concatenated feature space.***
>
> In more words, PARADE contains only two embeddings, one for the target class, and the other for sensitive attribute; we discard the sensitive attribute embedding at test time and solely use the target class. We incorporate a new loss term w.r.t. triplets mined by sensitive attribute label, which has not been done in any prior work to our knowledge. Finally, the intention of de-correlation in PARADE is *directional*: we want a target class embedding that is de-correlated from the sensitive attribute; whereas Milbich et al. use de-correlation to reduce feature redundancy in a concatenated embedding.
>
> On that note, we do want to highlight that we primarily leverage the general idea of joint multi-feature learning to learn a sensitive attribute embedding. While this was of primary focus in Milbich et al., multi-feature learning, as well as adversarial separation, is a generic approach that has seen previous usage already in e.g. [1, 2, 3].
>
> However, as per your suggestion, we have changed the wording to “a novel adaptation of previous zero-shot generalization techniques.”
>
> 2. > The authors show that the biases can be propagated to the downstream classification task and a naive method, training classifiers on the balanced dataset, cannot simply address the fairness issue. Since there are many bias mitigation methods ([1]) proposed for image classification, can authors also benchmark those bias mitigation methods for downstream classification? I understand that the authors want to pursue “fairness in representations” by proposing PARADE to address the fairness problem in the upstream embedding. However, I believe that the authors need to add some non-trivial bias mitigation methods to show that the bias cannot be fully addressed solely in the downstream stage, which can further demonstrate the necessity of achieving fairness in the upstream task, i.e., learning fairer representation in deep metric learning.
>
> Per your suggestion, we have run experiments with attribute-balanced sampling (which oversamples the minoritized subgroups, and which we title “Oversample”) and domain independent training, as these were the baseline and best-performing methods from [1], respectively. We have added comparison and analysis between these methods to Appendix Section C.1 and C.2 in our revision to supplement the results in the main paper, and added a pointer to the results in Section 6 of the main paper. See Tables 8 (CARS196), and 11 (CUB200) for overall performance and subgroup gaps downstream in baseline oversampling and domain independent training as detailed in [1]. We note that neither baseline oversampling nor domain independent training downstream with an “imbalanced embedding” improves over the “balanced” embedding downstream (for comparison, see downstream gaps of “balanced” embeddings in Table 7 for CARS196, Table 1 CUB200).
>
> Therefore, we conclude that these results further enforce our result that fairness cannot be fixed downstream for poorly constructed embedding spaces w.r.t. fairness. (For clarification, note that in our initial submission, the naive method of training downstream classifiers on a balanced dataset, was not done with oversampling or re-sampling of the minority groups, but with the original CARS196 / CUB200 datasets which are balanced; we contrived the imbalance used in the upstream embedding. We believe this constitutes best-case scenario, where the downstream classifiers are trained with real, balanced and non-repetitive data. We then compare this both upstream and downstream to the scenario where the upstream embedding also has access to the original balanced dataset.)
>
> [1] https://arxiv.org/abs/1909.11574
>
> [2] https://arxiv.org/abs/1801.04815
>
> [3] https://arxiv.org/abs/2004.05582

---

### Official Review · Reviewer_atFA · 2021-11-02

**Correctness:** 4
**Technical Novelty And Significance:** 3
**Empirical Novelty And Significance:** 3
**Recommendation:** 8
**Confidence:** 3

**Main Review:**

Strengths:
1. This is a well-written paper and it is easy to read.
2. This paper studies a new problem and provides a new perspective for the research in this field.
3. The experimental results are sufficient.

Weakness:

The proposed method looks simple and not novel enough (since the proposed method is largely based on previous work).

**Summary Of The Paper:**

This paper provides the finDML (fairness in non-balanced DML) benchmark in the scope of deep metric learning to characterize representation fairness and provide the method for reducing subgroup gaps in deep metric learning methods.


**Summary Of The Review:**

This paper is well-written and the proposed problem is intrersting. The supplementary materials provided by the author are very sufficient, and the comparative experiment is very detailed. The innovation of the method is general, but it systematically studies a new problem, so it is recommended to accept it.

---

> ### Author Response · Authors · 2021-11-19
> **PARADE explanation**
>
> Thank you very much for your positive response to the paper. We address the concerns here:
> > The proposed method looks simple and not novel enough (since the proposed method is largely based on previous work).
>
> We differentiate PARADE from previous fair representation learning methods in the last paragraph of Section 2. We re-iterate our discussion here: we focus on the main distinction that PARADE and our proposed fairness definitions offer DML-specific fairness boosting: the majority of disentangled representation learning methods (as cited in our paper) disentangle in a learned latent space. PARADE does not disentangle in a learned latent space, but rather in an embedding space intended to encode similarity. PARADE also learns similarity over the sensitive attribute itself, by producing a separate embedding, while the other works do not. Finally, PARADE does not remove all information about a sensitive attribute from the target class embedding, but rather modulates the de-correlation with the $\rho$ parameter.
>
> We have added additional differentiation between PARADE and the existing adversarial separation we used from Milbich et al. in the first paragraph of Section 4 in our revision. We include the added text here for reference:
>
> ***​​We enumerate several significant changes: 1) only target embedding released at test-time; 2) triplet formation and loss term w.r.t. sensitive attribute; 3) de-correlation with sensitive attribute as opposed to de-correlation to reduce redundancy in concatenated feature space.***
>
> In more words, PARADE contains only two embeddings, one for the target class, and the other for sensitive attribute; we discard the sensitive attribute embedding at test time and solely use the target class. We incorporate a new loss term w.r.t. triplets mined by sensitive attribute label, which has not been done in any prior work to our knowledge. Finally, the intention of de-correlation in PARADE is *directional*: we want a target class embedding that is de-correlated from the sensitive attribute; whereas Milbich et al. use de-correlation to reduce feature redundancy in a concatenated embedding.
>
> On that note, we do want to highlight that we primarily leverage the general idea of joint multi-feature learning to learn a sensitive attribute embedding. While this was of primary focus in Milbich et al., multi-feature learning, as well as adversarial separation, is a generic approach that has seen previous usage already in e.g. [1, 2, 3].
>
> [1] https://arxiv.org/abs/1909.11574
>
> [2] https://arxiv.org/abs/1801.04815
>
> [3] https://arxiv.org/abs/2004.05582

---

### Official Review · Reviewer_RQoJ · 2021-11-03

**Correctness:** 3
**Technical Novelty And Significance:** 2
**Empirical Novelty And Significance:** 3
**Recommendation:** 6
**Confidence:** 4

**Main Review:**

- First of all, the paper addresses an important problem, i.e., the fairness of the learned representation in deep metric learning. They provide an interesting observation that the existing deep metric learning methods become less fair when trained with class imbalance, and the learned bias propagates to the downstream tasks even when the downstream task datasets are balanced.

- They carefully designed the fairness measures in three different aspects, which are reasonable and help comprehensive understanding of the model behavior. In addition, they provide extensive experiments on five datasets with several DML methods that shows the decreased fairness for imbalanced dataset and motivates this work.

- I think the method itself serves as a good baseline but is technically not very novel. They employed the existing adversarial separation method and the idea to de-correlate two features are not new. Considering that there are two evaluation measures, accuracy and fairness, it would be interesting to study how to alleviate their trade-off.

- PARADE reduces the performace gaps between subgroups with different attributes on some datasets, but it does not always improve fairness depending on the dataset (Table 2).

- From the current explanation, it is unclear to me why "class" imbalance leads to the larger performance gaps between subgroups of different "attributes", which is also related to the proposed evaluation protocol.

- Since the proposed method can be appiled to any formulation of loss \mathcal{L}, it would be interesting to see if the method generalzes and improves fairness for different choices of loss formulations.

- As discussed in the limitations section, there is a trade-off between utility and the fairness. It is good that comparison of the actual accuracy is provided for some datasets in the supplmentary document. It shows that fairness was improved at the cost of overall low accuracy. It would be also interesting to discuss how to overcome the trade-off.

- Additional analysis of the effect of hyper-parameters (alpha and rho) would be necessary. It would be useful to discuss the strategy how to determine the values given a desired trade-off of fairness and utility.

- The method uses ground truth attribute labels to train a model. One simple baseline would be train a model with attribute-balanced sampling, i.e., to construct minibatch through balanced sampling over attributes. How does the method work on top of this baseline?

**Summary Of The Paper:**

This paper proposes three measures that evaluate the fairness of learned representations in multiple aspects. The authors empirically demonstrate that the existing metric learning approaches become less fair (i.e., shows larger performance gaps between attribute-based subgroups) when there is a class imbalance in the training data. Specifically, they propose a protocol that evaluates the fairness by manually adjusting the number of samples per class in the training set (finDML). In addition, they propose PARADE that de-correlates image embedding from auxiliary attibutes embedding via adversarial separation to improve fairness of the learned image embedding. In the experiments, the method shows overall improved fairness on five datasets.

**Summary Of The Review:**

Overall, the paper addresses an important problem, i.e., fairness in deep metric learning. Although the method does not always improve fairness compared to the baseline and the technical novelty seems weak, I think this work could motivate research on fairness for deep metric learning.

---
After authors' feedback:
I appreciate the extensive experiments and detailed answers. Most of my concerns are addressed.

---

> ### Author Response · Authors · 2021-11-19
> **Further PARADE explanation and Class Imbalance Experiments**
>
> Thank you for your comprehensive review of our paper, with important and constructive suggestions. We append additional experiments to our paper and address the listed concerns as follows:
> - > I think the method itself serves as a good baseline but is technically not very novel. They employed the existing adversarial separation method and the idea to de-correlate two features are not new. Considering that there are two evaluation measures, accuracy and fairness, it would be interesting to study how to alleviate their trade-off.
>
> We have added additional differentiation between PARADE and existing adversarial separation from Milbich et al. in the first paragraph of Section 4 in our revision. We include the added text here for reference:
>
> ***​​We enumerate several significant changes: 1) only target embedding released at test-time; 2) triplet formation and loss term w.r.t. sensitive attribute; 3) de-correlation with sensitive attribute as opposed to de-correlation to reduce redundancy in concatenated feature space.***
>
> In more words, PARADE contains only two embeddings, one for the target class, and the other for sensitive attribute; we discard the sensitive attribute embedding at test time and solely use the target class. We incorporate a new loss term w.r.t. triplets mined by sensitive attribute label, which has not been done in any prior work to our knowledge. Finally, the intention of de-correlation in PARADE is *directional*: we want a target class embedding that is de-correlated from the sensitive attribute; whereas Milbich et al. use de-correlation to reduce feature redundancy in a concatenated embedding.
>
> On that note, we do want to highlight that we primarily leverage the general idea of joint multi-feature learning to learn a sensitive attribute embedding. While this was of primary focus in Milbich et al., multi-feature learning, as well as adversarial separation, is a generic approach that has seen previous usage already in e.g. [1, 2, 3].
>
> Additionally, we differentiate PARADE from previous fair representation learning methods in the last paragraph of Section 2. We re-iterate our discussion here: we focus on the main distinction that PARADE and our proposed fairness definitions offer DML-specific fairness boosting: the majority of disentangled representation learning methods (as cited in our paper) disentangle in a learned latent space. PARADE does not disentangle in a learned latent space, but rather in an embedding space intended to encode similarity. PARADE also learns similarity over the sensitive attribute itself, by producing a separate embedding, while the other works do not. Finally, PARADE does not remove all information about a sensitive attribute from the target class embedding, but rather modulates the de-correlation with the $\rho$ parameter.
>
> Regarding the evaluation measures, we appreciate the suggestion and have added a new section to our appendix (Section C.5) that explores the tradeoff between utility and fairness in PARADE, over a grid of $\alpha_{SA}$ and $\rho$ values (see Figure 12). We have also incorporated additional discussion of the results in Figure 12 and pointers to further analysis in Section C.5 in the main paper at the following locations: last paragraph of Section 4; last paragraph of Section 7.
>
> - > From the current explanation, it is unclear to me why "class" imbalance leads to the larger performance gaps between subgroups of different "attributes", which is also related to the proposed evaluation protocol
>
> We clarify our scenarios here: we examine
> 1) Contrived class imbalance for a partitioning of classes where 50 classes are reduced in number of training samples (“minority classes”) and the remaining classes are increased in number of training samples (“majority classes”). The performance gaps are computed over the “minority classes,” and the “majority classes.” (CARS196 and CUB200)
> 2) Existing attribute imbalance where the distribution of training samples w.r.t. a sensitive attribute is not uniform (CelebA with Fitzpatrick Skintone, LFW with Race, and CUB200 with bird color). Here, the performance gaps are computed over subgroups partitioned by the sensitive attribute considered.
>
> [1] https://arxiv.org/abs/1909.11574
>
> [2] https://arxiv.org/abs/1801.04815
>
> [3] https://arxiv.org/abs/2004.05582

---

> ### Author Response · Authors · 2021-11-19
> **Additional PARADE experimental results**
>
> - > Since the proposed method can be appiled to any formulation of loss \mathcal{L}, it would be interesting to see if the method generalzes and improves fairness for different choices of loss formulations.
>
> Thank you for your suggestion. We have run additional experiments for PARADE with the Triplet loss for CUB200 (Table 2b in the revised main paper), and Arcface loss for the facial datasets (in appendix Section C for additional experiments). We include the markdown version of the tables here which provide additional support for PARADE results.
>
> CUB200 Table:
>
> |                           |           | PARADE (Triplet / Distance) | Triplet / Distance |
> |:-------------------------:|:---------:|:---------------------------:|:------------------:|
> |     Upstream Embedding    |  Recall@1 |        0.172 +- 0.027       |   0.195 +- 0.051   |
> |     Upstream Embedding    |    NMI    |        0.372 +- 0.291       |   0.359 +- 0.024   |
> |     Upstream Embedding    |    U_KL   |        0.174 +- 0.035       |   0.159 +- 0.018   |
> | Downstream Classification | Precision |        0.248 +- 0.038       |   0.308 +- 0.119   |
> | Downstream Classification |   Recall  |        0.276 +- 0.042       |   0.337 +- 0.123   |
> | Downstream Classification |  Accuracy |        0.148 +- 0.049       |   0.154 +- 0.029   |
>
> LFW Table:
>
> |                           |           | PARADE (Arcface) |     Arcface    |
> |:-------------------------:|:---------:|:----------------:|:--------------:|
> |     Upstream Embedding    |  Recall@1 |  0.039 +- 0.017  | 0.061 +- 0.017 |
> |     Upstream Embedding    |    NMI    |  0.048 +- 0.011  | 0.057 +- 0.003 |
> |     Upstream Embedding    |    U_KL   |  0.176 +- 0.019  | 0.157 +- 0.011 |
> | Downstream Classification | Precision |   0.04 +- 0.01   | 0.038 +- 0.012 |
> | Downstream Classification |   Recall  |  0.036 +- 0.018  | 0.041 +- 0.014 |
> | Downstream Classification |  Accuracy |  0.066 +- 0.017  |  0.076 + 0.015 |

---

> ### Author Response · Authors · 2021-11-19
> **Additional Results for Fairness / Utility Tradeoffs and Oversampling Experiments**
>
> - > As discussed in the limitations section, there is a trade-off between utility and the fairness. It is good that comparison of the actual accuracy is provided for some datasets in the supplmentary document. It shows that fairness was improved at the cost of overall low accuracy. It would be also interesting to discuss how to overcome the trade-off.
> - > Additional analysis of the effect of hyper-parameters (alpha and rho) would be necessary. It would be useful to discuss the strategy how to determine the values given a desired trade-off of fairness and utility.
>
> In response to both of the above bullet points, we added a new section to our appendix (Section C.5) that explores the tradeoff between utility and fairness in PARADE, over a grid of $\alpha_{SA}$ and $\rho$ values (see Figure 12). As stated above, we have also incorporated additional discussion of the results in Figure 12 and pointers to further analysis in Section C.5 in the main paper at the following locations: last paragraph of Section 4; last paragraph of Section 7. For reference, we include a portion of our added discussion here:
>
> ***In Figure 12, for Recall@1, we observe that at the location $\alpha_{SA} = 0.1$, $\rho = 500.$ in the optimization grid, PARADE reaches peak overall performance \textit{and} fairness (measured by low subgroup gap and high performance for the worst-performing subgroup) simultaneously. Thus, we could conclude that this choice of $\alpha_{SA}$ and $\rho$ represents an optimal tradeoff for utility and fairness in PARADE as measured by Recall@1. By the other displayed metrics, we see that $\alpha_{SA} = 0.1$, $\rho = 500.$ demonstrates a reasonable utility-fairness tradeoff. Therefore, the choice of  $\alpha_{SA} = 0.1$, $\rho = 500.$ would be optimal for PARADE in CUB200 bird color setting. Note that the choice of where to operate within this trade-off should depend on the application that is being targeted. For example, here we use Recall@1 to determine the optimal choice of hyperparameters and validate with the other two considered metrics. However, for LFW, which has a high population of singleton classes (see Figure 7), NMI would be a better metric to use for selecting optimal point.***
>
> - > The method uses ground truth attribute labels to train a model. One simple baseline would be train a model with attribute-balanced sampling, i.e., to construct minibatch through balanced sampling over attributes. How does the method work on top of this baseline?
>
> Per your suggestion, we have run experiments with attribute-balanced sampling (which we title “Oversample” for oversampling of the minoritized classes) and added comparison and analysis between the standard method with attribute-balanced sampling downstream to PARADE in Appendix Section C. Table 12b shows the overall performance and subgroup gaps for CUB200 bird color when standard deep metric learning losses are used upstream and attribute-balanced sampling is used for downstream classification (compare to Table 2b in main for PARADE results). Table 21b shows the overall performance and subgroup gaps for LFW Race attribute when standard deep metric learning losses are used upstream and attribute-balanced sampling is used for downstream classification (compare to Table 2a in main for PARADE results). Subgroup gaps in downstream classifiers of PARADE are significantly smaller than a baseline attribute-based sampling downstream. (CelebA not included as experiments were not finished in time but we intend to add these once the models are trained)

---

> ### Author Response · Authors · 2021-11-23
> **Any additional concerns?**
>
> Thank you again for your review! We wanted to follow up on our previous responses to ask if the reviewer has further questions about the paper, or the additional experimental results we have presented?

---

### Official Review · Reviewer_ryjh · 2021-11-04

**Correctness:** 3
**Technical Novelty And Significance:** 3
**Empirical Novelty And Significance:** 3
**Recommendation:** 6
**Confidence:** 3

**Main Review:**

Strengths:
------------
1) The first study that shows the effect of dml training on imbalanced datasets and how they are propagated to the downstream tasks
2) An exhaustive set of experiments showing this bias due to learning from imbalanced data
3) The paper is well written and organized
4) Reproducible as the code is available

Weaknesses:
----------------
1) Limited scope: The study proposed in this paper is limited to deep metric learning tasks and hence may not generalize to other tasks. The PARADE objective that employs DIVA technique to de-correlate the sensitive attribute from the discriminative features is also taylored to dml tasks. Hence all the analysis done in this paper would be limited to dml applications only. In addition, I have seen very few works that use dml for downstream tasks like classification because the main aim of dml is to learn a good metric (embedding space) where the features can be compared and generally used in scenarios where classification may not work well (large number of classes with few images in each class). I would like to know whether the analysis used here can be widely used for other tasks?

2) The method depends on user specified predefined attributes which could limit the applicability of the proposed approach. Also it can only be applied to datasets where these type of attributes are available.

**Summary Of The Paper:**

The paper presents a study on the effect of training dml techniques on imbalanced data and show the negative impact of learned representations on the downstream tasks. The fairness is analyzed through 3 properties of the representation space; a) inter-class alignment, b) intra-class alignment, c) uniformity showing that the bias in the upstream task (dml) is propagated to the downstream classification tasks even when the data for training the classifier (downstream task) is balanced. To address this, an objective (PARADE) that de-correlates the discriminative features from the sensitive attributes are learned during training. The experiments are conducted on 4 benchmark datasets.

**Summary Of The Review:**

The paper studies one of the sources of bias (data imbalance) in machine learning applications. Given the nature of current datasets and the bias with the current machine learning techniques, this is an important topic to explore. Although the proposed technique is currently applied/evaluated under restricted settings, this is one of the first work to do so for deep metric learning.

---

> ### Author Response · Authors · 2021-11-19
> **Fairness with awareness discussion and DML relevance**
>
> Thank you for your helpful comments, and constructive questions and feedback on our paper. We address the concerns in order as follows:
>
> 1. > Limited scope: The study proposed in this paper is limited to deep metric learning tasks and hence may not generalize to other tasks. The PARADE objective that employs DIVA technique to de-correlate the sensitive attribute from the discriminative features is also taylored to dml tasks. Hence all the analysis done in this paper would be limited to dml applications only. In addition, I have seen very few works that use dml for downstream tasks like classification because the main aim of dml is to learn a good metric (embedding space) where the features can be compared and generally used in scenarios where classification may not work well (large number of classes with few images in each class). I would like to know whether the analysis used here can be widely used for other tasks?
>
> Metric learning is at the basis of modern contrastive representation learning and extends to all kinds of visual similarity applications [1] and clustering. There has been consistent work recently in contrastively learned representation spaces embedded into large scale classification models [2]. Generally, deep metric learning models lend themselves well to zero- and few-shot transfer tasks, and are thus attractive to all manners of downstream tasks. Additionally, we note that we aim to define fairness in the representation space of deep metric learning methods (Section 3), not in any specific downstream task or use case (although the evaluation focuses on classification), so the analysis should extend to a diverse set of downstream applications or any deep metric learning related task. The analysis on downstream classification is largely due to the fact that fairness literature has focused primarily on classification and to motivate the fact that bias must be addressed in the training phase of the representation space for DML, not in later use cases of trained models.
>
> 2. > The method depends on user specified predefined attributes which could limit the applicability of the proposed approach. Also it can only be applied to datasets where these type of attributes are available.
>
> We agree that this is a limitation to the applicability of the proposed approach, as acknowledged in our discussion. This is a drawback of all “fairness with awareness” approaches that require the attribute information in order to enforce fairness constraints. We are encouraged that future work could consider PARADE where the attribute labels are determined by the recent idea of *computationally identifiable subgroups*, so no attribute information is required [3,4,5].
>
> [1] https://arxiv.org/abs/2002.08473
>
> [2] ​​https://arxiv.org/abs/2004.11362
>
> [3] https://arxiv.org/abs/1711.08513
>
> [4] https://arxiv.org/abs/1803.03239
>
> [5] https://arxiv.org/abs/2006.13114

---

> > ### Comment · Reviewer_ryjh · 2021-12-01
> > **Concerns addressed**
> >
> > Authors addressed some of my concerns during rebuttal and I agree with the authors about the scope of the work. Although the proposed method has some limitations (as listed by authors), it is the first paper that attempts to address fairness in deep metric learning. So I am changing the score to 7 (indicated 6 in the above ratings as 7 is not available)

---

### Decision · Program_Chairs · 2022-01-20

**Decision:**

Accept (Poster)

**Comment:**

This paper investigates an important problem, i.e., the fairness of the learned representation in deep metric learning, which is relatively under-explored by the research community. Observing that the existing metric learning approaches become less fair when trained on an imbalanced dataset, the authors propose finDML to benchmark previous methods on multiple imbalanced datasets with three newly proposed metrics.
Further, a PARADE module is adapted into this problem to tackle the fairness issue.

The paper is meticulously written of good structure, and well motivated by experimental findings. The authors have a deep and thorough discussion with reviewers, through which the mixed preliminary ratings became all positive, with most concerns well addressed. AC found no ground for rejection and thus recommended acceptance. Authors shall integrate all response material into the next revision.